# Safety and immunogenicity of Ad5-nCoV immunization after three-dose priming with inactivated SARS-CoV-2 vaccine in Chinese adults

Hangjie Zhang[1,8], Nani Xu[2,8], Yu Xu[3], Pan Qin[2], Rongrong Dai[4], Bicheng Xu[3], Shenyu Wang [1], Linling Ding[1], Jian Fu[1], Shupeng Zhang[3], Qianhui Hua[5], Yuting Liao[6], Juan Yang[6], Xiaowei Hu[2,9], Jianmin Jiang[1,4,7,9] & Huakun Lv [1,7,9] ✉

Data on the safety and immunity of a heterologous booster (fourth dose) after three-doses of inactivated SARS-CoV-2 vaccine in Chinese adults are limited. We evaluate the safety and immunogenicity of Ad5-nCoV in a randomized, double-blind, parallel-controlled phase 4 clinical trial in Zhejiang, China (NCT05373030). Participants aged 18–80 years (100 per group), administered three doses of inactivated SARS-CoV-2 vaccine ≥6 months earlier, are enrolled and randomized 1:1 into two groups, which are administered intramuscular Ad5-nCoV or inactivated SARS-CoV-2 vaccine (CoronaVac or Covilo). All observed adverse reactions are predictable and manageable. Ad5-nCoV elicits significantly higher RBD-specific IgG levels, with a geometric mean concentration of 2924.0 on day 14 post-booster, 7.8-fold that of the inactivated vaccine. Pseudovirus-neutralizing antibodies to Omicron BA.4/5 show a similar pattern, with geometric mean titers of 228.9 in Ad5-nCoV group and 65.5 in inactivated vaccine group. Ad5-nCoV booster maintains high antibody levels on day 90, with seroconversion of 71.4%, while that of inactivated vaccine is 5.2%, almost pre-booster levels. A fourth Ad5-nCoV vaccination following three-doses of inactivated SARS-CoV-2 vaccine is immunogenic, tolerable, and more efficient than inactivated SARS-CoV-2 vaccine. Ad5-nCoV elicits a stronger humoral response against Omicron BA.4/5 and maintains antibody levels for longer than homologous boosting.

Breakthrough infection cases of coronavirus disease 2019 (COVID-19) are a continuing issue in the clinical arena[1,2]. One potential reason is waning immunity accompanied by a significant decline in neutralizing antibodies (NAbs) 6 months after completion of the second or third dose of vaccine[3–6]. Another potential cause is the emergence of SARS-CoV-2 variants, called variants of concern (VOCs), that can escape immune attack, reducing the effectiveness of the vaccine. Omicron (B.1.1.529) especially, endowed with huge resistance to NAbs from

[1]Department of Immunization Program, Zhejiang Provincial Center for Disease Control and Prevention, Hangzhou 310057, China. [2]Xihu District Center for Disease Control and Prevention, Hangzhou 310007, China. [3]CanSino Biologics, Tianjin 300457, China. [4]School of Public Health, Hangzhou Medical College, Hangzhou 310053, China. [5]School of Medicine, Ningbo University, Ningbo 315211, China. [6]School of Public Health, Xiamen University, Xiamen 361005, China. [7]Key Lab of Vaccine, Prevention and Control of Infectious Disease of Zhejiang Province, Hangzhou 310057, China. [8]These authors contributed equally: Hangjie Zhang, Nani Xu. [9]These authors jointly supervised this work: Xiaowei Hu, Jianmin Jiang, Huakun Lv. ✉e-mail: hklv@cdc.zj.cn

either vaccinated or convalescent individuals, is making the prevention and control of the COVID-19 pandemic more difficult[7,8].

The administration of a fourth vaccine dose (additional booster) was recommended by the WHO for individuals aged ≥60 years, immunocompromised patients, and healthcare personnel after they have received the third dose[9]. Several population studies in Israel and Canada have shown the benefits of a fourth dose in reducing hospitalization, severe disease, and death, and protecting health systems[10–13]. One study showed that vaccination with either BNT162b2 or mRNA-1273 as the fourth dose increased IgG antibody and neutralizing antibody titers by a factor of 9–10[14]. Wang et al. demonstrated that a homologous booster of inactivated vaccine 6 months after the third dose further strengthened protective immune responses against both the ancestral SARS-CoV-2 strain and Omicron BA.2, although the peak antibody response to the receptor binding domain (RBD) was inferior compared with that after the third dose[15].

Immunity in those who had complete primary immunization was shown to be more efficiently restored by a heterologous booster than homologous boosters in clinical trials. In comparison to a third homologous dose of CoronaVac, administration of a recombinant adenoviral vectored vaccine, mRNA vaccine, or recombinant adenoviral-vectored ChAdOx1 nCoV-19 vaccine increased humoral and cellular immune responses[16–18]. Additionally, boosting ChAdOx1-primed adults with SCB-2019 (a protein subunit vaccine, S-Trimer) or mRNA vaccines induced higher levels of antibodies against a wild-type strain and SARS-CoV-2 variants than a homologous ChAdOx1 booster[19]. Giving the adenovirus-vectored vaccine booster to individuals vaccinated with inactivated vaccines can be highly beneficial, as Li et al. showed that administration of a heterologous booster with AD5-nCOV following initial vaccination with CoronaVac was more immunogenic than homologous boosting[20]. Until now, a large portion of the population of China is still vaccinated with three doses of inactivated vaccine, and the enhanced protection conveyed by a heterologous fourth booster dose has not been studied.

Here, we present the safety and immunogenicity results following heterologous booster immunization with Ad5-nCoV or homologous boosters with different inactivated SARS-CoV-2 vaccines (CoronaVac or Covilo) after three-dose priming with inactivated vaccine in healthy participants aged 18–80 years in a randomized, double-blind, parallel-controlled phase 4 trial. We further explored the potential benefits of anti-Omicron BA.4/5 protection, dynamic responses, and durations of antibody levels in individuals 3 months after the booster, as well as the relevant factors influencing the antibody responses.

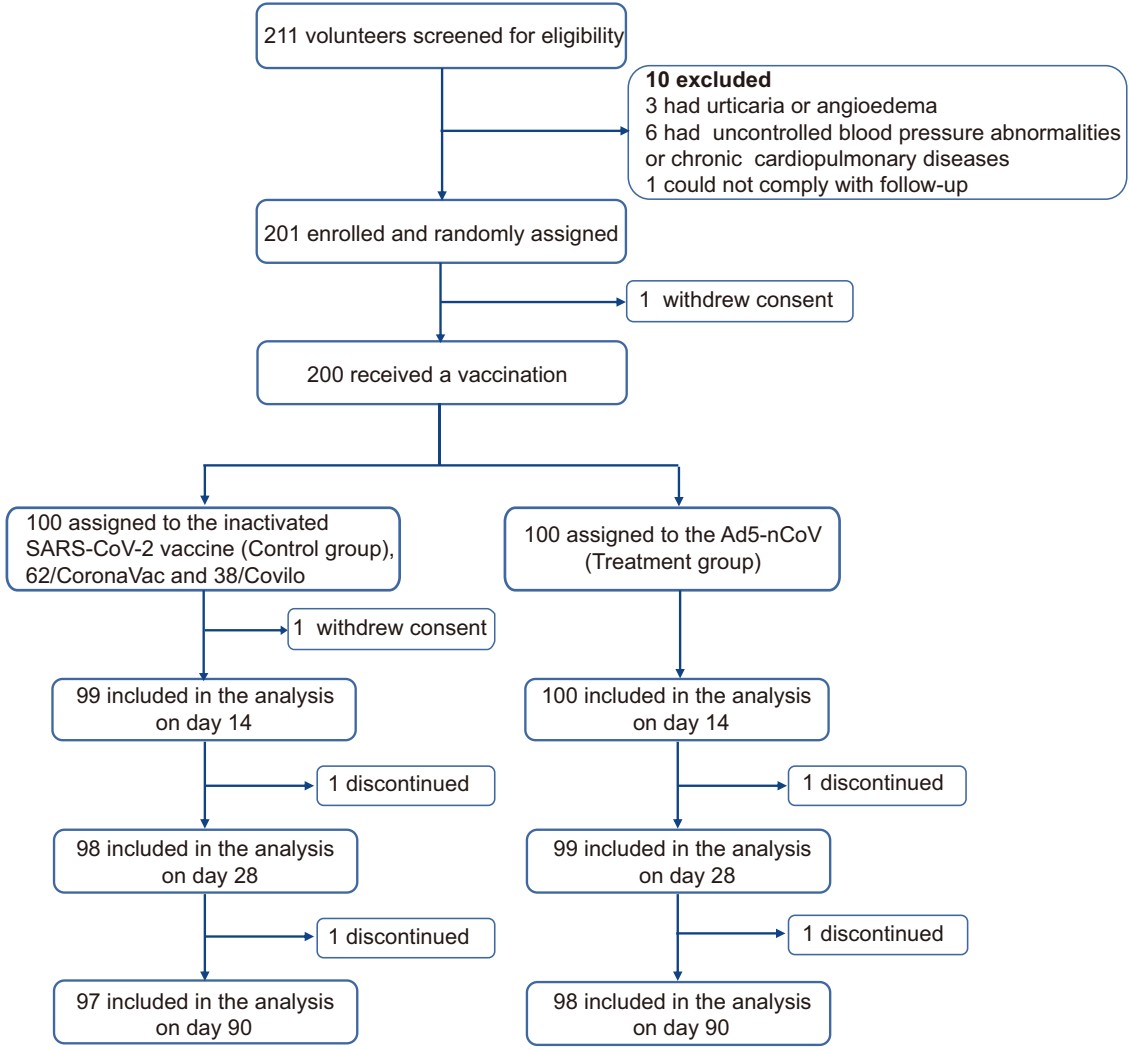

**Fig. 1 | Trial profile.** 211 volunteers were recruited and screened for eligibility, among which 201 participants were enrolled and randomly assigned; 200 participants received the booster dose vaccination, and 1 participant refused to receive a vaccination after randomization. Participants (98 and 97) in the two groups completed the planned visits within 90 days after vaccination. One subject discontinued the trial on day 14 after vaccination (primary safety included up to that point). Four participants discontinued the follow-up visits but provided safety records up to day 28.

**Table 1 | Baseline characteristics of enrolled participants**

| Variable[a] | Inactivated vaccine (n = 100)[b] | Ad5-nCoV (n = 100) |
|---|---|---|
| Sex | | |
| Male | 55 (55.0) | 55 (55.0) |
| Female | 45 (45.0) | 45 (45.0) |
| Age | | |
| 18–59 years | 59 (59.0) | 60 (60.0) |
| 60–80 years | 41 (41.0) | 40 (40.0) |
| Median age (IQR), years | 49 (38.0, 64.0) | 45.7 (37.3, 63.0) |
| Body mass index (kg/m²) | | |
| ≤18.4 | 1 (1.0) | 4 (4.0) |
| 18.5–24.9 | 65 (65.0) | 48 (48.0) |
| 25.0–29.9 | 30 (30.0) | 40 (40.0) |
| ≥30.0 | 4 (4.0) | 8 (8.0) |
| Time interval since the last priming dose of inactivated vaccine, months | | |
| Median (IQR) | 6.7 (6.3, 6.9) | 6.7 (6.3, 6.9) |
| Underlying chronic diseases[c] | | |
| Yes | 27 (27.0) | 32 (32.0) |
| No | 73 (73.0) | 68 (68.0) |

[a]Data are the number of participants (%) or median (IQR).
[b]62 participants received CoronaVac, and 38 participants received Covilo. Underlying chronic diseases included cardiovascular and cerebrovascular diseases, hypertension, and chronic obstructive pulmonary disease.

## Results

### Study participants
Between May 14 and 27, 2022, 211 volunteers aged 18–80 years who had received three doses of inactivated vaccine (CoronaVac or Covilo) ≥ 6 months earlier were recruited and screened for eligibility for this phase 4 trial. A total of 201 participants, including 120 adults (18–59 years) and 81 older individuals (60–80 years), were sequentially enrolled and randomly assigned to groups. One individual withdrew consent after randomization. In all, 200 participants received either a fourth dose of Ad5-nCoV (treatment group, n = 100) or inactivated vaccine (control group, n = 100) (Fig. 1). Finally, about 98 participants in the Ad5-nCoV group and 97 in the inactivated vaccine group completed the planned visits within 90 days after vaccination.

The mean age was 48 years (IQR, 38–64) for the whole study cohort, with 90 (45.0%) female participants and 59 (29.5%) with underlying chronic diseases. In total, 119 (59.5%) participants were aged 18–59 years (median age 39; IQR, 34–45), and 81 (40.5%) participants were aged 60–80 years (median age 64; IQR, 63–69) in the study. In both groups, the median time interval between the third dose of inactivated vaccine was 6.7 months (IQR 6.3, 6.9). At enrollment and before receiving the vaccine booster (day 0), the inactivated vaccine and Ad5-nCoV groups had similar GMT of anti-RBD-IgG values (63.1 vs 64.7) and seropositive rates (52.5% vs 55.0%). The baseline characteristics of the participants were comparable across the two groups (Table 1).

### Safety assessment
A total of 200 participants who received a fourth-dose booster were included in the safety analysis (Table 2). Within 14 days after boosting, a significantly higher frequency of solicited adverse reactions were reported by participants receiving Ad5-nCoV than participants receiving inactivated vaccine (55.0% vs 25.0%, P < 0.001) (Fig. 2). Participants receiving Ad5-nCoV reported more injection site (44.0% vs 18.0%, P < 0.001) and systemic adverse reactions (35.0% vs 12.0%, P < 0.001) than those receiving the inactivated vaccine. The incidence

**Table 2 | Solicited and unsolicited adverse reactions[a]**

| Variable | Inactivated vaccine (n = 100) | Ad5-nCoV (n = 100) | p |
|---|---|---|---|
| All solicited adverse reactions within 0–14 days[b] | | | |
| Total | 25 (25.0) | 55 (55.0) | <0.001 |
| Grade 1 | 22 (22.0) | 32 (32.0) | 0.111 |
| Grade 2 | 3 (3.0) | 13 (13.0) | 0.009 |
| Grade 3 | 0 (0.0) | 10 (10.0) | 0.001 |
| Injection site adverse reactions within 0–14 days | | | |
| Total | 18 (18.0) | 44 (44.0) | <0.001 |
| Pain | 16 (16.0) | 39 (39.0) | <0.001 |
| Induration | 3 (3.0) | 13 (13.0) | 0.121 |
| Redness | 3 (3.0) | 8 (8.0) | 0.121 |
| Swelling | 3 (3.0) | 11 (11.0) | 0.009 |
| Grade 3 | 0 (0.0) | 2 (2.0) | 0.497 |
| Rash | 1 (1.0) | 0 (0.0) | 1.000 |
| Itch | 2 (2.0) | 11 (11.0) | 0.010 |
| Systemic adverse reactions within 0–14 days[c] | | | |
| Total | 12 (12.0) | 35 (35.0) | <0.001 |
| Nausea | 0 (0.0) | 1 (1.0) | 1.000 |
| Fever | 2 (0.0) | 20 (20.0) | <0.001 |
| Grade 3 | 0 (0.0) | 7 (7.0) | 0.014 |
| Diarrhea | 3 (3.0) | 4 (4.0) | 1.000 |
| Arthralgia | 0 (0.0) | 12 (12.0) | <0.001 |
| Cough | 0 (0.0) | 5 (5.0) | 0.059 |
| Runny nose | 3 (3.0) | 4 (4.0) | 1.000 |
| Sneeze | 1 (1.0) | 2 (2.0) | 1.000 |
| Fatigue | 7 (7.0) | 25 (25.0) | 0.001 |
| Grade 3 | 0 (0.0) | 1 (1.0) | 1.000 |
| Headache | 3 (3.0) | 19 (19.0) | <0.001 |
| Appetite impaired | 1 (1.0) | 3 (3.0) | 0.621 |
| Oropharyngeal pain | 1 (1.0) | 4 (4.0) | 0.369 |
| Myalgia | 0 (0.0) | 10 (10.0) | 0.001 |
| Unsolicited adverse events within 28 days post vaccination | | | |
| Total | 7 (7.0) | 2 (2.0) | 0.170 |

Data are n (%). P-values of less than 0.05 were considered statistically significant.
[a]Solicited adverse reactions of maximum severity were recorded for each participant over the 14 days post-vaccination. Total refers to all participants with any grade adverse reaction or event. Adverse reactions and events were graded according to the scale issued by the China State Food and Drug Administration. Grade 3 = severe (i.e., prevented activity).
[b]There were no grade 4 solicited events reported in the trial.
[c]No events were reported for cellulitis, vomiting, or chest pain. Comparisons were analyzed by Fisher's exact test or Chi-squared test.

of unsolicited adverse events within 28 days after the vaccination was low in the two groups (2.0% vs 7.0%, P = 0.170). The most frequently reported solicited local adverse reaction was pain at the injection site (mostly grade 1), with an incidence of 16.0% in the inactivated vaccine group compared with 39.0% in the Ad5-nCoV group. Fatigue and fever were the most common solicited systemic adverse reactions, especially in the Ad5-nCoV group, with incidences of 25.0% and 20.0%, respectively.

Adverse reactions in the inactivated vaccine group were mild or moderate, with only 3.0% of participants having grade 2 injection site reactions (redness and swelling) and no grade 3 adverse reactions. However, the severity of adverse reactions was higher in Ad5-nCoV recipients, with 13.0% (grade 2) and 10.0% (grade 3) of participants. The symptoms of the two groups typically resolved within 2 days (IQR 1, 5). There were no grade 4 solicited events reported in the trial, and no thromboses, vaccine-related anaphylaxis, or other serious adverse events were documented during follow-up visits.

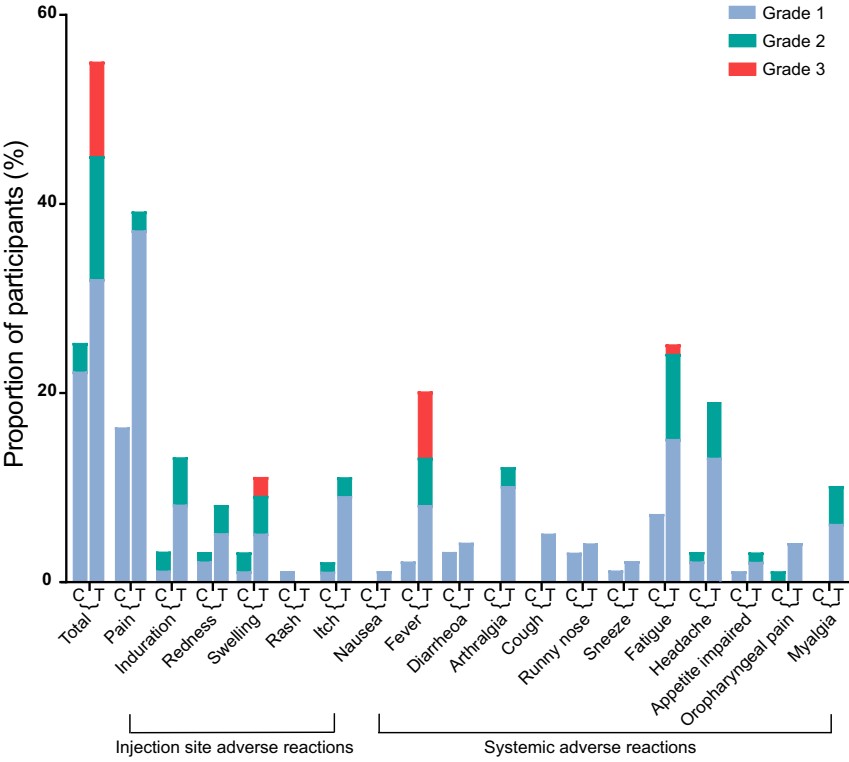

* No events were reported for cellulitis, vomiting, chest pain, etc.

**Fig. 2 | Solicited adverse reactions over 14 days post-vaccination.** Analysis was based on the safety cohort, which included all randomly assigned participants who received the booster vaccination. The maximum severity of solicited adverse reactions was recorded for each participant for each reaction. C = Inactivated SARS-CoV-2 vaccine group. T = Ad5-nCoV group. Source data are provided as a Source Data file.

Univariate analysis of influencing factors of solicited adverse reactions showed sex, age, and underlying chronic diseases had a significant impact ($P < 0.05$) on the frequency of local and systemic adverse reactions in both the Ad5-nCoV and inactivated vaccine group (Table 3). Participants 60–80 years of age had lower adverse reactions rates than those 18–59 years in both the inactivated vaccine (4.9% vs 39.0%, $P < 0.001$) and Ad5-nCoV (32.5% vs 70.0%, $P < 0.001$) groups. We used a logistic regression model of multivariate analysis to further explore the influencing factors and found being aged 60–80 years was the main protective factor against adverse reactions (OR = 0.12 for inactivated vaccine group; OR = 0.33 for Ad5-nCoV group).

**Immunogenicity assessment**
A total of 199 participants provided immunological samples following vaccination on day 0 and day 14. The GMT of RBD-specific binding IgG was 63.1 (95% CI 48.1–82.7) on day 0 and 263.3 (95% CI 191.0–292.4) on day 14 for the inactivated vaccine group (Table 4) and 64.7 (95% CI 50.7–82.7) on day 0 and 2250 (95% CI 1806.0–2803.0) on day 14 for the Ad5-nCoV group. The GMC of anti-RBD was 119.1 (95% CI 69.9–168.2) on day 0 and 333.3 (95% CI 247.8–418.8) on day 14 for the inactivated vaccine group and 105.2 (95% CI 77.5–132.8) on day 0 and 2924.0 (95% CI 2305.0–3542.0) on day 14 for the Ad5-nCoV group. Heterologous boosting with Ad5-nCoV elicited a significantly higher RBD-specific IgG antibody concentration (7.8-fold, IQR 3.6, 21.7; $P < 0.0001$) than homologous boosting with the inactivated vaccine (Fig. 3a). Seropositivity was achieved in 93 of 99 participants (93.9%, 95% CI 87.3–97.7) in the inactivated vaccine booster group and 100% of the 100 participants (95% CI 96.4–100.0) in the Ad5-nCoV booster group on day 14 after vaccination (Fig. 3b). Seroconversion (4-fold increase in GMT of RBD-specific binding IgG) was measured in 37 of 99 participants (37.4%, 95% CI 27.9–46.7) in the inactivated vaccine group, which

was significantly fewer participants ($P < 0.0001$) than in the Ad5-nCoV group (91.0%, 95% CI 83.6–95.8) (Fig. 3c). Similarly, boosting with Ad5-nCoV significantly increased RBD-specific IgG antibody responses with a GMFI of 34.8 (95% CI 26.5–45.7) compared with the inactivated vaccine with a GMFI of 3.7 (95% CI 3.0–4.6) from day 0 to 14 ($P < 0.0001$) (Fig. 3d). The GMFI of the Ad5-nCoV/ inactivated vaccine was 9.5 (95% CI 7.0–12.9) on day 14, which met the criteria for non-inferiority (the lower bound of the confidence interval was >0.67).

The GMT of BA.4/5 pseudovirus-neutralizing antibody were 34.1 (95% CI 31.0–37.4) on day 0 and 65.5 (95% CI 58.1–76.0) on day 14 for the inactivated vaccine group, and 30.0 (95% CI 27.2–33.1) on day 0 and 228.9 (95% CI 187.4–279.5) on day 14 for the Ad5-nCoV group (Table 4). In line with the RBD-specific binding antibody levels, both cohorts showed increases in neutralizing antibody titers against the pseudovirus BA.4/5 variant compared with baseline, with GMFI of 2.0 (95% CI 1.7–2.3) and 7.6 (95% CI 6.3–9.3), respectively. Booster vaccination with Ad5-nCoV more potently elicited BA.4/5 neutralizing antibody levels than the inactivated vaccine ($P < 0.0001$), with a 3.3-fold increase (IQR 1.9, 6.2) (Fig. 3d). Additionally, seropositivity was significantly higher in the Ad5-nCoV group (79.0%, 95% CI 69.7–86.5) than the inactivated vaccine group (34.3%, 95% CI 25.1–44.6), and there was a similar difference in seroconversion rates (74.0% vs 17.2%, $P < 0.001$).

We explored the dynamic responses and duration of antibody levels after booster vaccinations with Ad5-nCoV or inactivated SARS-CoV-2 vaccine. The levels of RBD-specific binding IgG and BA.4/5 pseudovirus-neutralizing antibodies dropped from day 28 after the booster in both groups (Fig. 3). On day 90, the GMC of RBD-specific IgG antibody was 152.7 (95% CI 104.7–200.7) in the inactivated vaccine group, with a GMFI 1.7-fold higher (95% CI 1.4–2.1) than baseline, and 1275.0 (95% CI 868.3–1681.0) in the Ad5-nCoV group, with a 9.1-fold

**Table 3 | Analysis of factors influencing solicited adverse reactions**

| | Inactivated vaccine (n = 99[a]) | | | | Ad5-nCoV (n = 100) | | | |
|---|---|---|---|---|---|---|---|---|
| | Adverse reactions rates % | Chi-Squared test | Logistic test | | Adverse reactions rates % | Chi-Squared test | Logistic test | |
| | | p | OR | 95% CI | | p | OR | 95% CI |
| Sex | | 0.023 | | | | 0.034 | | |
| Male | 9 (16.4) | | ref | | 25 (45.5) | | ref | |
| Female | 16 (36.4) | | 1.74 | 0.61–4.90 | 30 (66.7) | | 2.20 | 0.91–5.32 |
| Age | | <0.001 | | | | <0.001 | | |
| 18–59 years | 23 (39.0) | | ref | | 42 (70.0) | | ref | |
| 60–80 years | 2 (4.9) | | 0.12 | 0.03–0.56 | 13 (32.5) | | 0.33 | 0.11–0.94 |
| Body mass index (kg/m²) | | 0.158 | - | - | | 0.157 | - | - |
| ≤18.4 | 0 (0.0) | | | | 4 (100.0) | | | |
| 18.5–24.9 | 20 (30.8) | | | | 31 (57.4) | | | |
| 25.0–29.9 | 5 (16.7) | | | | 15 (44.1) | | | |
| ≥30.0 | 0 (0.0) | | | | 5 (62.5) | | | |
| Underlying chronic diseases | | 0.016 | | | | 0.001 | | |
| Yes | 2 (7.7) | | ref | | 10 (31.3) | | ref | |
| No | 23 (31.5) | | 0.39 | 0.08–2.02 | 45 (66.2) | | 0.48 | 0.16–1.48 |
| Third dose of Inactivated vaccine | | 0.262 | - | - | | 0.050 | - | - |
| CoronaVac | 18 (29.0) | | | | 42 (61.8) | | | |
| Covilo | 7 (18.9) | | | | 13 (40.6) | | | |

Logistic regression model (Enter method) was used to conduct multivariate analysis to further explore the influencing factors of solicited adverse reactions rates. The inclusion criterion α = 0.05, and the exclusion criterion α = 0.10. Calculated the OR and its 95% CI for "Yes" versus "no". Test level α = 0.05. P-values of less than 0.05 were considered statistically significant.
[a]Not including one participant who did not provide information of solicited adverse reactions within 0–14 days.

higher GMFI (95% CI 7.0–11.8). Seropositivity and seroconversion decreased to 76.3 (95% CI 66.6–84.3) and 5.2 (95% CI 1.7–11.6), respectively. Fortunately, seropositivity (100.0%, 95% CI 96.3–100) and seroconversion (71.4%, 95% CI 61.4–80.1) rates were maintained high levels for longer durations for the Ad5-nCoV booster. For neutralizing activity to the pseudovirus BA.4/5 variant, the inactivated vaccine group GMT was 31.9 (95% CI 27.7–36.8), with a GMFI 1.0-fold higher (95% CI 0.8–1.1) than baseline; and the Ad5-nCoV group GMT was 99.8 (95% CI 82.8–120.2), with a 3.3-fold higher GMFI (95% CI 2.7–4.0). Seropositivity (8.2%, 95% CI 3.6–15.6) and seroconversion (0.0%, 95% CI 0.0–3.7) on day 90 were lower in the inactivated vaccine group than seropositivity (61.2%, 95% CI 50.8–70.9) and seroconversion (38.8%, 95% CI 29.1–49.1) in the Ad5-nCoV group after the booster vaccination.

Additionally, we explored the immunogenicity of a sub-group given a booster vaccination with Ad5-nCoV. The results showed that there was no statistic difference in antibody responses of Ad5-nCoV booster on those with a prime vaccination of CoronaVac or Covilo (P > 0.05) (Fig. 4, Supplementary Table 1, 2, 3 and 4, Supplementary Fig. 1).

**Factors influencing antibody levels**

The seropositivity and seroconversion results for the BA.4/5 pseudovirus-neutralizing antibody grouped by age (i.e., 18–59 years and 60–80 years), sex, BMI, chronic conditions, and adverse reactions are presented in Table 5. The seropositivity rates (44.1% for 18–59 years and 20.0% for 60–80 years, P = 0.013) and GMT (74.3 for 18–59 years and 56.4 for 60–80 years, P = 0.045) for the Omicron BA.4/5 neutralizing antibody were significantly different between the different age groups of participants receiving the inactivated vaccine and on day 14 after booster vaccination, but there were no significant differences among participants receiving Ad5-nCoV (P = 0.193, P = 0.670). Importantly, the influence of age on Omicron BA.4/5 neutralizing antibody was less apparent in the Ad5-nCoV group than the inactivated vaccine group, especially in the early days following booster vaccination (Fig. 5a). The incidence of adverse reactions significantly influenced seropositivity and GMT of Omicron BA.4/5

neutralizing antibody in both groups. Participants with adverse reactions exhibited a higher seropositivity (56.0% vs 27.0%, P = 0.008) and GMT (82.7 vs 61.7, P = 0.060) in the inactivated vaccine group, and higher seropositivity (89.1% vs 66.7%, P = 0.001), seroconversion (85.5% vs 60.0%, P = 0.004) and GMT (304.9 vs 161.2, P = 0.001) in the Ad5-nCoV group. Relatively severe adverse reactions, such as grade 3 reactions, may have elicited more potent and durable antibodies after booster vaccination, especially for participants receiving Ad5-nCoV (Fig. 5b). There were no significant differences found for sex, BMI, chronic condition groups or third dose of inactivated vaccine.

## Discussion

Vaccination provides immune-protection against SARS-CoV-2, owing to the induction of neutralizing antibodies and cellular immunity. However, as we are faced with the rapid evolution of SARS-CoV-2 variants worldwide and the virus's increasing immune escape ability[21,22], a fourth booster dose is being incorporated into many vaccination schedules[10,13]. In this study, we investigated the safety and immunogenicity of a heterologous booster (Ad5-nCoV) or homologous booster (CoronaVac or Covilo inactivated vaccine) in healthy participants aged 18–80 years who were previously immunized with three doses of the inactivated vaccine for over 6 months. Data in this study showed that the participants' humoral responses were rapidly and strongly elevated by the Ad5-nCoV heterologous fourth-dose booster on day 14 after immunization, not only against WT SARS-CoV-2 but also against the Omicron BA.4/5 variant. Homologous boosting with the inactivated SARS-CoV-2 vaccine, however, elicited weaker antibody responses, especially in terms of neutralizing activity to Omicron BA.4/5. This phenomenon was similar to the higher antibody responses induced by a third booster dose of a heterologous vaccine than those induced by homologous vaccines[17,23].

The antibody levels following homologous booster of inactivated vaccine in participants decayed faster than with a heterologous Ad5-nCoV booster. In a study into the use of a third-dose Ad5-nCoV or inactivated vaccine, Jin et al. demonstrated that orally administering aerosolized Ad5-nCoV following two-dose CoronaVac priming was

**Table 4 | RBD-specific IgG antibodies and pseudovirus-neutralizing antibodies to Omicron BA.4/5 before and after booster vaccination**

| Variable | Inactivated vaccine | | | | Ad5-nCoV | | | |
|---|---|---|---|---|---|---|---|---|
| | Day 0 (n = 99[a]) | Day 14 (n = 99[a]) | Day 28 (n = 98[a]) | Day 90 (n = 97[a]) | Day 0 (n = 100) | Day 14 (n = 100) | Day 28 (n = 99[a]) | Day 90 (n = 98[a]) |
| **Anti-RBD-IgG** | | | | | | | | |
| GMT | 63.1 (48.1–82.7) | 263.3 (191.0–292.4) | 170.5 (138.6–209.8) | 106.7 (83.7–135.9) | 64.7 (50.7–82.7) | 2250 (1806.0–2803.0) | 1594 (1284.0–1979.0) | 993.2 (786.4–1254.0) |
| GMC (RU/mL) | 119.1 (69.9–168.2) | 333.3 (247.8–418.8) | 234.4 (182.6–286.2) | 152.7 (104.7–200.7) | 105.2 (77.5–132.8) | 2924.0 (2305.0–3542.0) | 2142.0 (1649.0–2635.0) | 1275.0 (868.3–1681.0) |
| Seropositive rate (%) | 52.5 (42.2–62.7) | 93.9 (87.3–97.7) | 90.8 (83.2–95.7) | 76.3 (66.6–84.3) | 55.0 (44.7–65.0) | 100.0 (96.4–100.0) | 100.0 (96.3–100.0) | 100.0 (96.3–100.0) |
| Seroconversion rate (%) | NA | 37.4 (27.9–46.7) | 27.6 (19.0–37.5) | 5.2 (1.7–11.6) | NA | 91.0 (83.6–95.8) | 88.9 (81.0–94.3) | 71.4 (61.4–80.1) |
| GMFI | NA | 3.7 (3.0–4.6) | 2.7 (2.2–3.4) | 1.7 (1.4–2.1) | NA | 34.8 (26.5–45.7) | 25.0 (19.0–32.8) | 9.1 (7.0–11.8) |
| **Neutralizing antibodies to Pseudovirus (BA.4/5)** | | | | | | | | |
| GMT | 34.1 (31.0–37.4) | 65.5 (58.1–76.0) | 51.7 (45.1–59.2) | 31.9 (27.7–36.8) | 30.0 (27.2–33.1) | 228.9 (187.4–279.5) | 162.1 (131.7–199.6) | 99.8 (82.8–120.2) |
| Seropositive rate (%) | 4.0 (1.1–10.0) | 34.3 (25.1–44.6) | 26.5 (18.1–36.4) | 8.2 (3.6–15.6) | 1.0 (0.02–5.4) | 79.0 (69.7–86.5) | 74.7 (65.0–82.9) | 61.2 (50.8–70.9) |
| Seroconversion rate (%) | NA | 17.2 (10.3–26.1) | 8.2 (3.6–15.5) | 0.0 (0.0–3.7) | NA | 74.0 (64.3–82.3) | 58.6 (48.2–63.4) | 38.8 (29.1–49.2) |
| GMFI | NA | 2.0 (1.7–2.3) | 1.1 (0.9–1.3) | 1.0 (0.8–1.1) | NA | 7.6 (6.3–9.3) | 5.4 (4.4–6.6) | 3.3 (2.7–4.0) |

Data are mean or geometric mean (95% CI) or n (%).
GMT geometric mean titer, GMC geometric mean concentration, GMFI geometric mean fold increase.
[a]Discontinued: subjects failed to provide blood and therefore were not included in the antibody assay.

persistently more immunogenic than a third CoronaVac dose 12 months after the second dose[24]. Other studies showed that the duration of the antibody response to a homologous boost with a third dose of inactivated SARS-CoV-2 vaccine was unsatisfactory, although the booster restored antibody levels quickly and even raised levels above those induced by the two priming doses[25,26]. Our data herein revealed that, in a scheme involving a fourth dose following three doses of inactivated vaccine, there was a greater decline in antibody responses against WT SARS-CoV-2 and Omicron BA.4/5 elicited by the inactivated vaccine on day 90 after the booster, whereas Ad5-nCoV maintained comparatively high levels. Wang et al. demonstrated that the peak antibody response to the RBD after the fourth dose of an inactivated vaccine was inferior to that induced by the third dose[15]. Even though we have no data on the RBD-specific antibody levels after the third dose to verify this phenomenon, our results suggested the fourth-dose Ad5-nCoV booster further elevated the peak value following three doses of the inactivated SARS-CoV-2 vaccine.

Previous clinical trials reported that most adverse events after Ad5-nCoV were mild or moderate in severity and might be associated with cellular immunity reactivity[20,27]. In these trials, 81% of participants reported at least one adverse reaction, and 9% had Grade3 adverse reactions after the first Ad5-nCoV vaccination; whereas 34.4% of participants reported adverse reactions and 2.1% were severe after heterologous boosting with Ad5-nCoV following two doses of inactivated SARS-CoV-2 vaccine. Similarly, our data indicated that adverse reactions resulting from heterologous boosting with Ad5-nCoV after three-dose priming with an inactivated SARS-CoV-2 vaccine were predictable and manageable, although they presented with a higher incidence than those induced by homologous boosting with an inactivated vaccine. Tsang et al. demonstrated that incidences of adverse reactions decreased with the frequency of vaccination[28]. This may also be a result of the monitoring sensitivity, as adverse reactions were reported in 25.3% of participants that received inactivated SARS-CoV-2 vaccine in our study, a higher percentage than seen in other homologous booster studies[29]. Interestingly, each of the factors (male, aged 60–80 years, and chronic diseases) is associated with a lower frequency of local and systemic adverse reactions.

The morbidity and mortality rates in older adults increase after infection with SARS-CoV-2, and an effective COVID-19 vaccine is urgently needed[30,31]. Our previous study showed that older adults tend to have a weaker immune response than younger adults after receiving an inactivated SARS-CoV-2 vaccine[6]. Importantly, heterologous boosting with Ad5-nCoV increased antibody levels, irrespective of age, especially in the early stages after booster vaccination. This phenomenon was previously seen in a study showing that a third-dose Ad5-nCoV booster increased antibody levels regardless of age[32]. Vaccine reactogenicity is characterized by inflammatory responses of the humoral and cellular immune systems and can result in vaccine-related adverse reactions[33]. There are therefore concerns about the correlations between adverse effects and antibody responses following COVID-19 vaccinations. Participants with adverse reactions exhibited higher antibody responses when given boosters of Ad5-nCoV or inactivated SARS-CoV-2 vaccine, and grade 3 adverse reactions may elicit more potent and durable antibodies.

This study had some limitations. First, live virus in vitro neutralization tests against WT SARS-CoV-2 and Omicron BA.4/5 were not conducted in our study, and these need to be further explored, although RBD-specific binding IgG and pseudovirus-neutralizing antibody levels induced by vaccination were shown to correlate with live virus neutralizing antibody levels in our previous study[6]. Second, the screening of participants' histories of SARS-CoV-2 infection was only based on an epidemic surveillance system, and we did not conduct SARS CoV-2 nucleic acid and serology tests at baseline. Third, although we only evaluated the antibody responses induced by vaccination, and not the cellular immunity, it is expected that a heterologous Ad5-nCoV

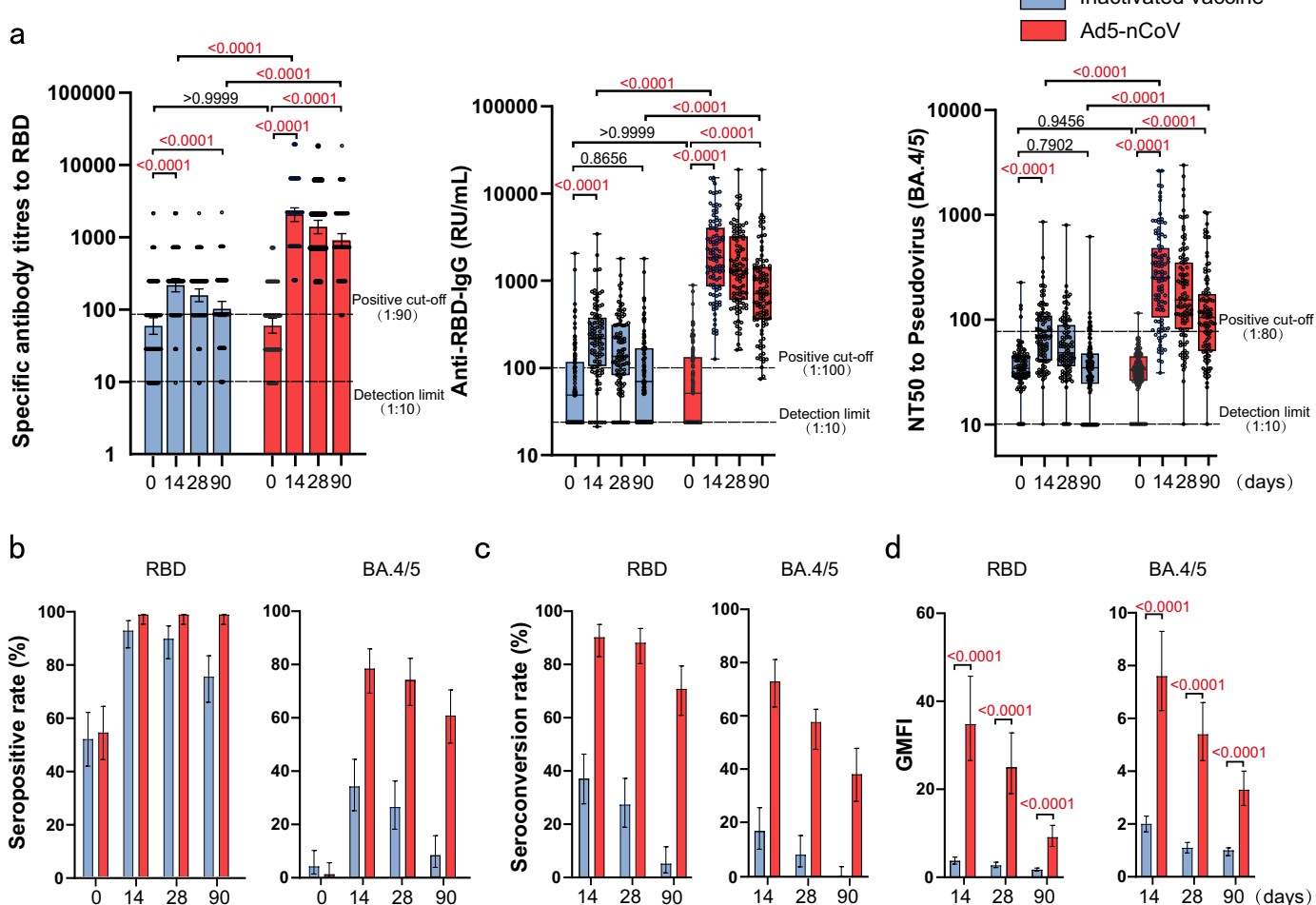

**Fig. 3 | Antibody responses before and after booster vaccination. a** GMT and GMC of SARS-CoV-2 RBD-specific IgG antibodies, and GMT of pseudovirus-neutralizing antibodies to Omicron BA.4/5 on day 0 (before booster vaccination) and days 14, 28, and 90 (after booster vaccination) in Ad5-nCoV (*N* = 100, 100, 99, 98) and inactivated SARS-CoV-2 vaccine groups (*N* = 99, 99, 98, 97).
**b**–**d** Seropositive rates (%) (**b**), seroconversion rates (%) (**c**), and GMFI (**d**) of RBD-specific antibodies and pseudovirus-neutralizing antibody 14, 28, and 90 days after booster compared with baseline. Data were shown as Geometric mean ± 95% CI or as box and whiskers in (**a**), indicating median (middle line), 25th, 75th percentile

(box), mean ± 95% CI in (**b**–**d**). Seroconversion was defined as at least a 4-fold increase in antibody titers at different time points after booster compared with baseline (day 0). NT50 = 50% neutralization titer. Statistical significance was determined by two-tailed Student's *t*-test and one-way ANOVA with Tukey's multiple comparisons test. *P* < 0.05 was considered statistically significant and marked with red color. 95% CI: 95% confidence interval, GMT geometric mean titer, GMC geometric mean concentration, GMFI geometric mean fold increase. Source data are provided as a Source Data file.

booster would enhance the cellular immune response more than a homologous booster of inactivated vaccine[20]. Fourth, the study was conducted on a seronegative population and no longer reflects the state of the current population. However, as a strategy for COVID-19 vaccination, our results suggest heterologous boosting is more effective than homologous boosting for enhancing antibody levels, which is also applicable to individuals who have been infected. Finally, the study was an immunogenicity evaluation that does not translate to a difference in protection against hospitalizations or severe disease. There has been no direct correlation found between immunization and protection or duration of protection[34]. Additionally, we did not evaluate the actual levels of protection from infection by current and emerging variants conveyed by the vaccines, and these need to be monitored with real-world observational studies.

In conclusion, our study provided evidence that a fourth dose of Ad5-nCoV following a three-dose inactivated SARS-CoV-2 vaccination is more immunogenic, tolerable than a fourth inactivated SARS-CoV-2 vaccine. Ad5-nCoV elicits a stronger humoral response and restores higher peak and more durable antibody levels than the inactivated SARS-CoV-2 vaccine. Nevertheless, while Ad5-nCoV seems to be highly

potent and protective against Omicron lineages, next-generation vaccines may be needed to provide better protection against the emergence of highly transmissible SARS-CoV-2 variants capable of immune escape.

## Methods
### Study design and participants
We performed a single-center, randomized, double-blind, parallel-controlled trial of a fourth vaccine dose with heterologous booster (Ad5-nCoV) or homologous booster (inactivated SARS-CoV-2 vaccine) in a population that had completed a three-dose inactivated SARS-CoV-2 vaccine regime in Hangzhou, Zhejiang Province, China. Healthy participants aged 18–80 years with stable medical conditions were recruited from the community and divided into groups of younger adults (aged 18–59 years) and older adults (aged 60–80 years) at a 2:1 ratio. All participants had received three doses of inactivated vaccine (the third dose was CoronaVac or Covilo) 6 months before the screening visit.

The exclusion criteria included a history of laboratory-confirmed SARS-CoV-2 infection (based on the epidemic

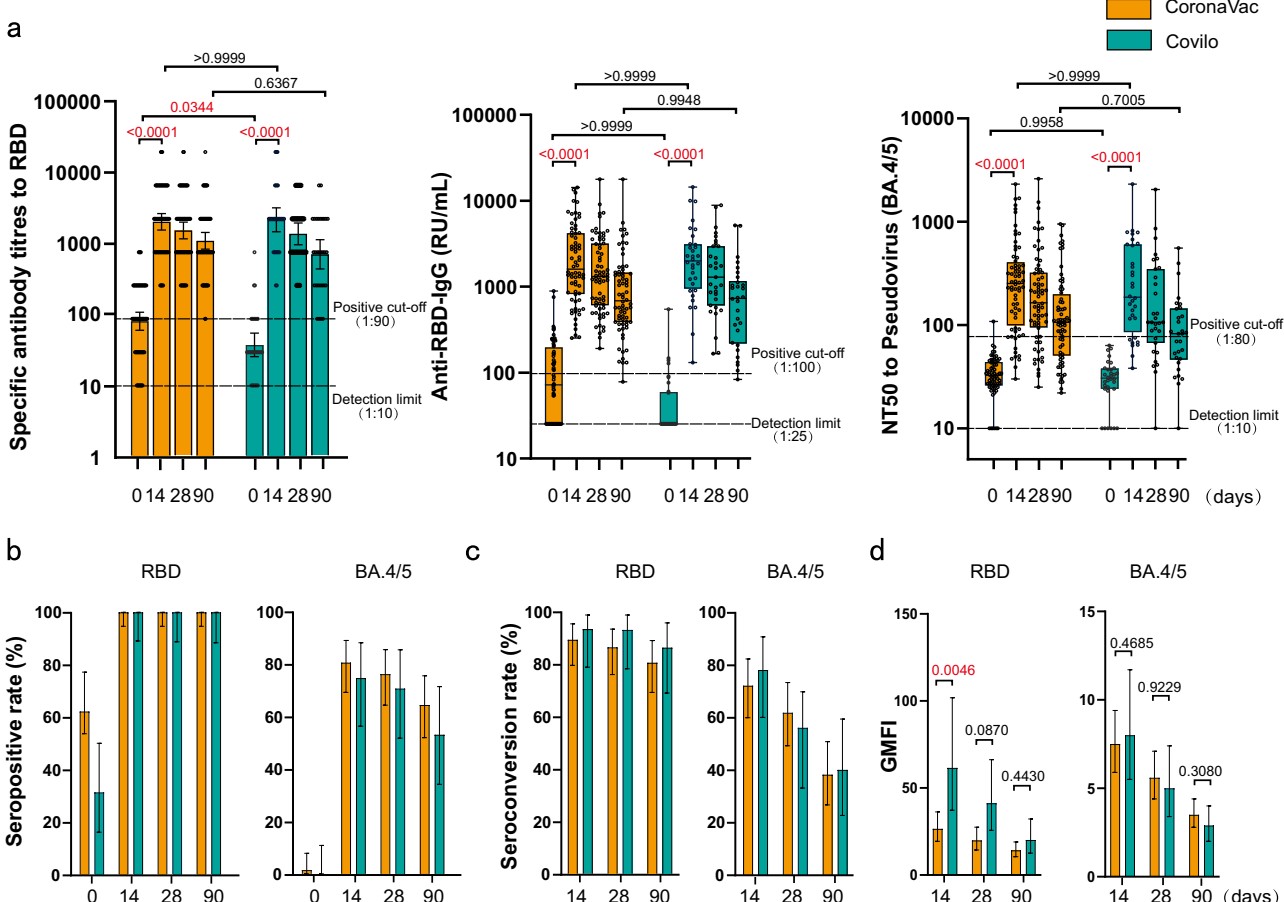

**Fig. 4 | Antibody responses of sub-group given booster vaccination with Ad5-nCoV. a** GMT and GMC of SARS-CoV-2 RBD-specific IgG antibodies, and GMT of pseudovirus-neutralizing antibodies to Omicron BA.4/5 on day 0, 14, 28, and 90 in sub-group given CoronaVac ($N = 68, 68, 68, 68$) or Covilo ($N = 32, 32, 32, 31$). **b–d** Seropositive rates (%) (**b**), seroconversion rates (%) (**c**), and GMFI (**d**) of RBD-specific antibodies and pseudovirus-neutralizing antibody 14, 28, and 90 days after booster compared with baseline. Data were shown as Geometric mean ± 95% CI or as box and whiskers in (**a**), indicating median (middle line), 25th, 75th percentile (box), mean ± 95% CI in (**b–d**). Seroconversion was defined as at least a 4-fold increase in antibody titers at different time points after booster compared with baseline (day 0). NT50 = 50% neutralization titer. Statistical significance was determined by two-tailed Student's *t*-test and one-way ANOVA with Tukey's multiple comparisons test. $P < 0.05$ was considered statistically significant and marked with red color. 95% CI: 95% confidence interval, GMT geometric mean titer, GMC geometric mean concentration, GMFI geometric mean fold increase. Source data are provided as a Source Data file.

information system in China); pregnant or lactating women; new onset of fever; an immunosuppressive condition; previous vaccination with any other authorized vaccine within 28 days before screening (or the intention to receive such a vaccine within 14 days after the booster dose); history of severe adverse reactions (e.g., anaphylaxis) to adenovirus-vectored vaccines or coronavirus vaccines; administration of (or the intention to receive) any blood, plasma, or immunoglobulin products within 90 days before screening; participants with severe and/or uncontrolled cardiovascular disease, respiratory disease, hypertension (systolic pressure ≥180 mmHg/ diastolic pressure ≥110 mmHg), diabetes, or neurologic illness; and any special circumstances that researchers considered may increase the risk of individuals participating in the study or interfere with the evaluation of the initial objectives of the study.

All participants signed the informed consent form before enrollment. The trial protocol was reviewed and approved by the Research Ethics Committee of the Zhejiang Provincial Center of Disease Control and Prevention, and no changes were made after the initiation of the study (ethics code number: 2022-021-01). The trial was prospectively registered with https://clinicaltrials.gov/ (NCT05373030) and conducted following the principles of the Declaration of Helsinki, ICH Good Clinical Practice guidelines.

### Vaccines

Trial vaccines included CoronaVac (Vero Cell; Beijing Sinovac Research & Development, Beijing, China) and Covilo (Vero Cell; Beijing Institute of Biological Products Co., Beijing, China), which are inactivated whole virion vaccines to wild-type SARS-CoV-2 with aluminum hydroxide as the adjuvant, administered intramuscularly at 0.5 mL per dose. Convidecia (CanSino Biologics, Tianjin, China) is a replication-defective Ad5-vectored vaccine expressing the full-length spike gene of wild-type SARS-CoV-2 (WuhanHu-1) supplied as a liquid formulation with a concentration of $5 \times 10^{10}$ viral particles per 0.5 mL.

### Randomization and masking

The block randomization method was used with block sizes of 2 and stratified by age group. Eligible participants in each cohort were randomly assigned at a 1:1 ratio to receive Ad5-nCoV or inactivated vaccine. Randomization was done using a sealed enveloped system integrated with the electronic case report forms in the Open Clinica platform. A random number was assigned to each participant by allocation of the next available randomized entry in the randomization list (generated by an independent statistician using SAS (version 9.4)) that was established before the start of the study.

During the double-blind phase, the participants, investigators, and outcome assessors were blinded. The personnel who prepared

**Table 5 | Analysis of factors influencing antibody levels (Omicron BA.4/5)**

| Variable | Inactivated vaccine (n = 99) | | | | | | Ad5-nCoV (n = 100) | | | | | |
|---|---|---|---|---|---|---|---|---|---|---|---|---|
| | Seropositivity rates % | p | Seroconversion rates % | p | GMT (95% CI) | p | Seropositivity rates % | p | Seroconversion rates % | p | GMT (95% CI) | p |
| Sex | | 0.219 | | 0.851 | | 0.127 | | 0.786 | | 0.748 | | 0.557 |
| Male | 16 (29.1) | | 9 (16.4) | | 60.6 (51.5–71.3) | | 44 (80.0) | | 40 (72.7) | | 216.8 (168.1–279.7) | |
| Female | 18 (40.9) | | 8 (17.8) | | 74.7 (59.6–93.6) | | 35 (77.8) | | 34 (75.6) | | 244.5 (176.4–338.9) | |
| Age | | 0.013 | | 0.108 | | 0.045 | | 0.193 | | 0.780 | | 0.670 |
| 18–59 years | 26 (44.1) | | 4 (9.8) | | 74.3 (61.7–89.6) | | 50 (83.3) | | 45 (75.0) | | 265.7 (204.1–345.9) | |
| 60–80 years | 8 (20.0) | | 13 (22.0) | | 56.4 (47.0–67.7) | | 29 (72.5) | | 29 (72.5) | | 183.0 (134.9–248.2) | |
| Body mass index (kg/m²) | | 0.090 | | 0.535 | | 0.084 | | 0.157 | | 0.421 | | 0.218 |
| ≤18.4 | 0 (0.0) | | 0 (0.0) | | 48.0 | | 4 (100.0) | | 4 (100.0) | | 395.0 (139.3–1120) | |
| 18.5–24.9 | 25 (39.1) | | 12 (18.5) | | 73.3 (61.9–86.8) | | 40 (74.1) | | 37 (68.5) | | 230.7 (174.5–305.1) | |
| 25.0–29.9 | 9 (30.0) | | 5 (16.7) | | 59.9 (47.0–76.3) | | 29 (85.3) | | 26 (76.5) | | 188.3 (140.7–252.1) | |
| ≥30.0 | 0 (0.0) | | 0 (0.0) | | 33.3 (22.8–48.5) | | 6 (75.0) | | 7 (87.5) | | 378.3 (107.5–1331) | |
| Underlying chronic diseases | | 0.159 | | 0.807 | | 0.020 | | 0.230 | | 0.740 | | 0.013 |
| Yes | 6 (23.1) | | 5 (18.5) | | 51.1 (38.8–67.2) | | 23 (71.9) | | 23 (71.9) | | 159.3 (115.2–220.2) | |
| No | 28 (38.4) | | 12 (16.4) | | 73.0 (62.8–85.0) | | 56 (82.4) | | 51 (75.0) | | 271.5 (212.3–347.1) | |
| Adverse reactions | | 0.008 | | 0.877 | | 0.060 | | 0.001 | | 0.004 | | 0.001 |
| Yes | 14 (56.0) | | 4 (16.0) | | 82.7 (63.5–107.9) | | 49 (89.1) | | 47 (85.5) | | 304.9 (237.9–390.7) | |
| No | 20 (27.0) | | 13 (17.3) | | 61.7 (52.9–72.1) | | 30 (66.7) | | 27 (60.0) | | 161.2 (119.0–218.3) | |
| Third dose of Inactivated vaccine | | 0.757 | | 0.136 | | 0.567 | | 0.501 | | 0.519 | | 0.731 |
| CoronaVac | 22 (35.3) | | 12 (19.7) | | 68.6 (56.8–82.8) | | 55 (80.9) | | 49 (72.1) | | 234.5 (183.8–299.1) | |
| Covilo | 12 (32.4) | | 3 (8.3) | | 63.1 (52.7–75.6) | | 24 (75.0) | | 25 (78.1) | | 217.4 (150.4–314.3) | |

Comparisons were analyzed by Fisher's exact test or Chi-squared test, Cochran–Armitage test were applied to analyze categorical data. P-values of less than 0.05 were considered statistically significant.

and administered the vaccinations were not blinded, but they were not otherwise involved in other trial procedures or data collection and signed a confidentiality agreement. In addition, analyses of the safety and immunogenicity outcomes were carried out in a blinded manner by the lab technicians.

## Procedures

In May 2022, 200 eligible participants received one dose of Ad5-nCoV or inactivated vaccine via intramuscular injection. The variety of the fourth dose of inactivated vaccine was the same as that used for the pre-received third dose. After the booster vaccination, all participants were monitored for 30 min for any immediate adverse reactions and instructed to record adverse events for the next 14 days on a participant diary card. Solicited injection site events included pain, redness, swelling, induration, itching, and cellulitis, while systemic events included fever, malaise, muscle ache, joint pain, fatigue, nausea, and headache. Unsolicited adverse events within 28 days were also reported by the participants and recorded. Serious adverse events (SAEs) self-reported by participants were documented throughout the study. Adverse events were graded as mild (grade 1), moderate (grade 2), severe (grade 3), or life-threatening (grade 4), according to the scale issued by the China State Food and Drug Administration (version 2019), and the causality with immunization was estimated before unmasking.

To assess the presence of anti-SARS-CoV-2 antibodies, a 10-ml blood sample was collected from each participant at baseline before they received the booster dose and 14, 28, and 90 days after receiving the booster dose. The quantitative measurement of wild-type SARS-CoV-2 RBD-specific IgG responses was performed using the commercial Anti-SARS-CoV-2 RBD-IgG ELISA kit (Vazyme Medical Technology, Nanjing, China). The lower limit of detection was 1:10, and values below this limit were imputed and defined as 1:10. The positive cutoff values for RBD-specific IgG antibodies were defined as titers of 1:90[34]. The titer was converted to relative units per milliliter (RU/ml) with reference to the WHO international standard for anti-SARS-CoV-2 immunoglobulin (NIBSC code 20/136). The lower limit of detection was 10 RU/ml, and values below this limit were imputed and defined as 10 RU/ml. The positive RBD-specific IgG response was defined as a concentration ≥100 RU/ml. We also assessed the levels of neutralizing antibody against the Omicron BA.4/5 subvariant using a pseudovirus-neutralization test (a vesicular stomatitis virus pseudovirus system that expresses the spike glycoprotein, 50% neutralization titer [NT50]) using the Reed-Muench method with a positive cutoff NAb titer ≥1:80[35]. The lower limit of detection was 1:10, and values below this limit were imputed and defined as 1:10. Seroconversion was defined as at least a four-fold increase in antibody levels over the baseline values.

## Outcomes

The primary outcomes were the occurrence of solicited local or systemic adverse reactions and assessments of geometric mean concentration (GMC)/geometric mean titer (GMT), seroconversion/seropositive rates, and geometric mean fold increase (GMFI) of antibodies that bind SARS-CoV-2-specific RBD and pseudovirus-neutralizing antibodies against Omicron BA.4/5 on day 14 post-booster dose. Safety secondary outcomes included information on unsolicited adverse events for 28 days after immunization and serious adverse events recorded throughout the whole follow-up visits. Immunological secondary outcomes included the above indices of humoral immune responses to vaccination at 28 and 90 days after vaccination. The exploratory outcomes were analyses of influencing factors (including age, sex, BMI, and chronic diseases) to adverse reactions and immunogenicity.

## Statistical analysis

The sample size calculation was based on the assumption that heterologous booster immunization with Ad5-nCoV after three-dose

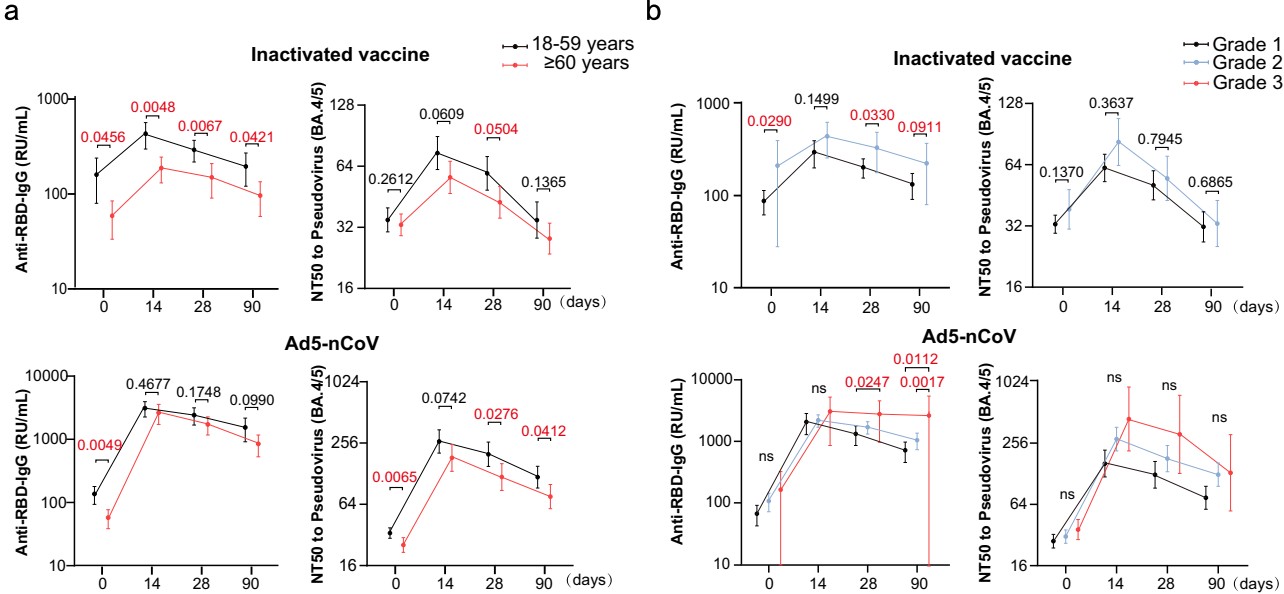

**Fig. 5 | Influence of age and adverse reaction grade on antibody levels. a, b** GMC of SARS-CoV-2 RBD-specific IgG antibodies and GMT of pseudovirus-neutralizing antibodies to Omicron BA.4/5 in different age groups (18–59 years and 60–80 years) (**a**) and adverse reaction grade groups (**b**) on day 0, 14, 28, and 90 after booster vaccination with Ad5-nCoV ($N = 100, 100, 99, 98$) or inactivated SARS-CoV-2 vaccine ($N = 99, 99, 98, 97$). Data were shown as geometric mean ± 95% CI.

Statistical significance was determined by two-tailed Student's $t$-test and one-way ANOVA with Tukey's multiple comparisons test. ns no significance; $P < 0.05$ was considered statistically significant and marked with red color. 95% CI 95% confidence interval, GMT geometric mean titer, GMC geometric mean concentration, GMFI geometric mean fold increase. Source data are provided as a Source Data file.

priming with inactivated SARS-CoV-2 vaccine would elicit non-inferior and superior concentrations of neutralizing antibodies compared with a homologous booster dose with inactivated SARS-CoV-2 vaccine. We assumed that the one-sided α of inspection level was 0.025, the ratio of Ad5-nCoV group to inactivated SARS-CoV-2 vaccine group was 1:1, the non-inferiority limit of the GMT ratio was 0.67 (after 10 is the bottom log = −0.174), and the actual GMT ratio of the two groups was 2; thus, the study needed to recruit 80 participants in each group to achieve 99.86% power. When the sample size of each group was 80, the superiority of the GMT level of the Ad5-nCoV group was greater than that of the inactivated SARS-CoV-2 vaccine group, and the estimated power of the test was 88.16%, which met the test requirements. Thus, we decided on a total sample size of 200, with 100 for each group after adjusting for an attrition rate of 20% due to loss to follow-up. Power Analysis and Sample Size software (LLC, USA version 11.0.7) was used in these calculations.

Sex, age, BMI, and other clinical characteristics were collected for each vaccine recipient. We used the medians and interquartile ranges (IQR) for age and time interval, and numbers (percentages) for categorical variables. All participants who received the booster dose were included in the safety population (Safety set [SS]). Safety data are presented as counts and percentages of participants with at least one solicited (local or systemic) adverse event. The analysis was based on the intention-to-treat cohort and calculated with chi-squared test or Fisher's exact test. The immunogenicity objectives are reported based on the per-protocol set (PPS), and missing data were not imputed. Participants experienced no major protocol violations; complied with all inclusion criteria/exclusion criteria; completed the vaccinations within the time window, as required in the protocol; and completed all blood samplings. Levels of antibodies against SARS-CoV-2 are presented as the GMC/GMT, seroconversion/seropositive rate, and GMFI with 95% CI. The 95% CI was calculated based on the t-distribution of the log-transformed values back-transformed to the original scale. We used Student's $t$-test and one-way ANOVA with Tukey's multiple comparisons test to analyze the log-transformed antibody titers and categorical data. The Wilcoxon rank-sum test was used to analyze data

not following a normal distribution, and the chi-squared test, Fisher's exact test or the Cochran–Armitage test were applied to analyze categorical data. A logistic regression model (enter method) was used to conduct multivariate analysis. $P$-values of less than 0.05 were considered statistically significant (*$P < 0.05$; **$P < 0.01$, ***$P < 0.001$, ****$P < 0.0001$). All statistical analyses were conducted with SPSS 18.0 (IBM Corporation, Armonk, NY, USA), SAS Statistical Software v.9.4 (SAS Institute Inc., Carey, NC, USA), R (version 4.2.1) and GraphPad Prism 9 (San Diego, CA, USA).

## Data availability

The raw data on demographics and clinical status of participants, are protected and not available due to data privacy laws. The processed data are available by specific request to the corresponding author Huakun Lv(hlv@zj.cdc.cn). Source data are provided with this paper.

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

## Acknowledgements

We thank all the generous volunteer subjects who enrolled in the study. This work was supported by the Key Research and Development Program of Zhejiang Province (2021C03200, J.J., H.L., H.Z.); the Key Program of Health Commission of Zhejiang Province/Science Foundation of National Health Commission (WKJ-ZJ-2221, H.L., H.Z.); the Major Program of Zhejiang Municipal Natural Science Foundation (LD22H190001, J.J., H.Z.); the Explorer Program of Zhejiang Municipal Natural Science Foundation (LQ23H100001, H.Z.). CanSino Biologics contributed to the study design and clinical data collection; CanSino Biologics coauthors revised and approved the final version of the manuscript.

## Author contributions

H.L., J.J., and H.Z. were the principal investigators who designed and performed the research and coordinated the study; N.X., X.H., P.Q., Q.H., Y.L., and J.Y. led and participated in the site work, including the recruitment, follow-up, and data collection; S.W., J.F., L.D., Y.X., S.Z. and B.X. supervised the study; H.Z., Q.H., Y.L., and J.Y. were responsible for laboratory analyses; H.Z. and R.D. did the statistical analysis and wrote the manuscript. X.H., H.L., and J.J. interpreted the data and revised the manuscript. All authors critically reviewed the manuscript and approved the final version.

## Competing interests

Y.X., S.Z., and B.X. are employees of CanSino Biologics and contributed to the conceptualization of the study (clinical protocol and electronic case report form design) but did not participate in the analysis or interpretation of the data presented in the manuscript. All other authors declare no competing interests.
