## [Peer Review File · Nature Communications]

Safety and immunogenicity of Ad5-nCoV immunization after three-dose priming with inactivated SARS-CoV-2 vaccine in Chinese adultsReviewers' Comments:

Reviewer #1:

Remarks to the Author:

This manuscript compared a 4th dose of inactivated versus Ad5 SARS-CoV-2 vaccine in a population of Chinese adults. The results are important for the large population who have received exclusively inactivated vaccines in China, and show a benefit to the Ad5 vaccine. The methods are strong, the results are valid and the discussion of the results is well-balanced. The manuscript would benefit from a better focus on the main points, as it is quite long. The exclusion on the basis of prior infection is unclear to me, and an important factor in interpreting the results.

Introduction. The introduction is long and contains much generic information on the pandemic. The entire first paragraph, for example, could be omitted.

On page 4, lines 77-79 be more specific about the heterologous boosters. From looking at the references, there were Ad5 and mRNA vaccines.

Methods. Please provide additional detail on the vaccines, including the manufacturer. I am not familiar with the Covilo vaccine.

Page 5, line 100. How was a history of laboratory-confirmed SARS-CoV-2 determined?

Page 7, line 133. Why did the variety of the fourth dose of the inactivated vaccine be the same as that used for the third dose? Shouldn't these be interchangeable? Is this practical to implement in a public health program?

line 148. How were the anti-SARS-Co-V antibody tests used, and what was the specific test? Were persons excluded based on this test?

Statistical analysis, line 175. I assume you mean the sample size was 200, or the study was "designed to include" 200?

Define the per-protocol set.

Results.

Page 10, line 208. Please specify months.

Page 11, lines 233-237. This is an interpretation of the results and should be moved to the discussion section.

Page 13, lines 285 to end of paragraph. Likewise, this is a conclusion and should be in the discussion section.

Discussion. For the most part, it is balanced and well-written. One page 18, I am confused by the mention of screening of histories. The serostatus of an individual and prior infection are important predictors of response to vaccine (those with prior infection generally having more robust responses). I can't follow how this was determined - through history and antibody testing? Or only history? And how were participants excluded?

The graphs are well-done. Table 2 is redundant and not needed in my view.

Reviewer #2:

Remarks to the Author:

The Zhang et al paper is a tour de force of vaccine trial design, execution, data reporting and analysis! This study provided a critical comparative analysis of inactivated vaccine and adenovirus-based vaccine as the fourth dose following three doses of inactivated vaccine. Evaluation of safety and

antibody responses were clearly stated and presented, showing adenovirus-based vaccine had higher immune response and relatively high reactogenicity comparing to the homologous inactivated vaccines. Furthermore, the authors analyzed the relationship of antibody response with age and reactogenicity. The study has great implication for people who have received 3 doses of inactivated vaccine and the data can't be clearer than presented here. Although additional evaluation of immunogenicity could be done, the authors already pointed out in the limitation of the study. And I applaud that the authors did the nAb response against BA.4/5 rather than the more convenient ancestral strain. Overall, the study is sound and solid with great significance. Kudos to the authors for the great work!

Reviewer #3:

Remarks to the Author:

This is an important clinical trial that presents both safety and immunogenicity results against SARS-CoV-2 after a heterologous booster dose with Ad5-nCoV after three-dose priming with inactivated SARS-CoV-2 vaccine in Chinese adults. The data are of very good quality, and are presented in a well written manuscript, providing important and interesting results to add to the current body of evidence on heterologous COVID-19 booster vaccinations.

However, I do have some comments on the manuscript.

Hypothesis testing/analysis comments:

- The manuscript states there was no sample size calculation. How was sample size determined if no formal sample size was performed? Was the sample size pre-determined? Can the authors please clarify this, as there are these inconsistencies between the manuscript and the protocol: both the protocol and the study SAP contain a power calculation – is this post-hoc, or based on pre-determined sample size?
- The protocol and SAP both show that the main hypothesis under evaluation was non-inferiority of Ad5-nCoV to inactivated vaccine control, based on geometric mean ratio (GMR) of the GMT for the two arms (for the anti-RBD IgG results, judging by the values in the SAP section 2.3). This manuscript contains no GMR results and therefore doesn't address the hypotheses as stated in the protocol. The anti-RBD day 14 GMR with CI needs to be provided, as planned in the SAP. The non-inferiority comparison needs to be made to then justify the superiority comparison. I would suggest that the GMR should be adjusted for baseline (pre-booster) levels of anti-RBD. Otherwise, a reason for not performing the primary hypothesis needs to be stated in the text and justified.
- There is no information on the numbers of participants who received CoronaVac or Covilo as their third dose (and therefore as their trial dose) – please include this information in both the study flow diagram (Figure 1) and baseline characteristics table (Table 1). I would suggest the authors to provide a supplementary table of control group GMTs split by CoronaVac and Covilo recipients, so readers can see similar results between these sub-groups (justifying pooling both vaccines into one 'inactivated vaccine' control group).
- I would suggest for the authors to perform an exploratory immunogenicity sub-group analysis by prime vaccination schedules (CoronaVac and Covilo), to observe if there are differences in the effect of Ad5-nCoV booster on those with a prime series of CoronaVac vs Covilo, with adjustments made for differences in participant populations. Potentially obvious limitations are sample size, but it would still be good to perform this exploratory analysis to observe if there are any detectable differences.
- Can the authors please justify their choice in using Kruskal-Wallis tests? These non-parametric tests are performed on log-transformed data, which should be normally distributed. Please consider using a parametric test to analyse this data.
- Can the authors please comment on SAEs – this is a secondary endpoint but has not been mentioned in the results section. If no SAEs occurred then please state this. Could the authors please also justify why SAEs were only collected for the first 28 days post-vaccination?

Population comments:

- 200 participants received a vaccination on the trial, but the safety population only includes 199 participants. Line 176-177 states "all participants who received the booster dose were included in the safety population". Why is this one participant missing from the safety analysis? Please explain this in the text and in Figure 1. If this is the participant that discontinued then did they discontinue from all safety follow-ups immediately after vaccination? (If not then there should be data, and if so then isn't this a withdrawal rather than a discontinuation?)
- Was there an upper limit on recruitment age (80 years)? The inclusion criteria states ≥ 18 years but further details in the SAP and protocol show an upper limit of 80 years. If so, please make this clear in the manuscript.
- Can the authors please explain the justification for their 2:1 ratio for 18-59 and ≥ 60 years age groups. On what basis was this chosen, particularly given that the authors state a fourth dose is recommended by the WHO only to those ≥ 60 years?
- The immunogenicity analysis is stated to have been performed on the per-protocol dataset. The protocol and SAP contain definitions for this, but they do not state if blood draws needed to be within the pre-specified windows (as in the protocol, section 8.8.1). Please include a clear definition of the per-protocol dataset in the manuscript.

Randomisation comments:

- The protocol states that block randomisation was used. Please include details of this in the manuscript, with details of block sizes used.
- Randomisation was stratified by age group – please include this detail in the 'Randomization and Masking' section of the manuscript.
- Was randomisation stratified by sex as well as age? If so, please also mention this in the randomisation section.

Discussion comments:

- Can the authors please include in their discussion how the results of this trial will translate to the current population, given that the study is on a seronegative population but this no longer reflects the state of the current population.
- The authors show a difference in anti-RBD IgG between the 2 age groups in the inactivated vaccine group, but not in participants receiving Ad5-nCoV. They commented that the same results were not seen when observing neutralising antibody. This is an interesting finding – please can the authors further comment on this in the discussion.
- Can the authors please include in the discussion how their findings might translate to protection against infection. This is mentioned as a limitation but it needs further discussion. Many published studies show a difference in vaccines that doesn't translate to a difference in protection against hospitalisations or severe disease.
- Related to the point above, on line 387 please remove the statement 'more efficient'. This implies that Ad5-nCoV has higher vaccine efficacy compared to inactivated vaccine as fourth dose, but there is no direct evidence for this. Although immunogenicity is clearly higher in the Ad5-nCoV arm, this doesn't necessarily translate to more protection against severe disease/hospitalisation (e.g. seen in the following article, when comparing vaccine efficacy of BNT162b2 and CoronaVac in the population: <https://www.sciencedirect.com/science/article/pii/S1473309922003450?via=ihub>)
- Please can the authors add to their statement in their limitations section of not looking at cellular immune data – although cellular outcomes weren't looked at, please comment on what would be expected based on the current literature on cellular response after Ad5-nCoV vaccination.
- Line 324-325 – the authors state the "Ad5-nCoV booster also dramatically increased the decay time compared to the homologous inactivated vaccine", however the rate of decay isn't a result looked at in this manuscript and there are only results up to 90 days following vaccination. Do the authors mean that the antibody levels at day 90 following booster vaccination were significantly higher in participants receiving Ad5-nCoV than those receiving inactivated vaccine? Please can the authors change their wording here to help clarify.
- Lines 338-341 – it's stated that "our results showed the fourth-dose Ad5-nCoV booster further elevated the peak value following three doses of the inactivated SARS-CoV-2 vaccine". Please change

“showed” to “suggested” – as the authors already state they have no data on peak value following three doses to show this.

- Line 386 – please change “safe” to ‘tolerable’, for example. “Safe” cannot be stated in this small sample, given that there is insufficient power to detect rare safety events (and SAEs were only collected for 28 days following trial vaccination).

Other comments:

- Can the authors please clarify what is meant by “duration of antibody levels” (lines 86, 267)? Is this duration above a certain threshold, or above pre-vaccination levels?
- Line 36 – please change statement “maintained antibody levels for longer” to “maintained higher antibody levels at day 90”. Or instead please clarify that they are talking about levels above the seropositive threshold.
- Line 102 – can the authors please clarify if “receipt of any vaccine within 30 days before or after each study dose” should refer to each dose prior to the study, not the on-study dose? Otherwise this criteria overlaps with the next (lines 103-104) and is confusing.
- Did any of the assays have lower or upper limits of detection (definitely appears that anti-RBD IgG assay had lower limit of detection)? If so, please include in the text how these were dealt with in the immunogenicity analysis (i.e. were values below or above limits of detection imputed?).
- Please mention the different inactivated vaccines in lines 29 (abstract) and 83-84 (introduction) – it is not mentioned until line 123 currently.
- Please can the authors change their use of the word “robust” in this manuscript (lines 40, 316, 388)? The use of “robust” is subjective and unclear.
- I would suggest that seroconversion is defined in line 165 (it is not defined until line 241).
- What tests were used to produce the p-values stated for the safety analysis (lines 215, 217, 219)? Please include this in the Statistical Analysis section after line 178 (all following information regarding statistical tests in this section relates only to the immunogenicity analysis).
- Line 196 is the only time it’s mentioned that this is a phase 4 trial. Suggest including this earlier in the manuscript, e.g. in line 91.
- The text should state that no grade 4 solicited events were reported in the trial. Please also note this in Table 2, at least in the footnote.
- Many additional solicited adverse reactions were collected but not reported in the manuscript. Although this is likely due to these reactions not occurring, it is still important to relay this information. Please add to the footnote on Table 2 stating that no events were reported for cellulitis, vomiting, chest pain, etc. (stating all the solicited adverse reactions collected, for which no events were reported). Please add a similar footnote to Figure 2.
- Differences in sex, age, chronic diseases were explored in the relationship between vaccine and safety events, but no adjustment for multiple testing was used. Could the authors please justify this? Please consider adjusting for multiple testing. Same comment applies for looking at factors influencing antibody levels. (Table 4 and table 5.)
- Line 238 – please remove the word ‘Approximately’.
- Please provide units for GMTs in both the text and for Table 3.
- Suggest that lines 354-356 should be made clearer. Currently it reads as though a combination of these 3 factors are associated with lower frequency of adverse reactions, as opposed to each of these factors individually.

Figure and table comments:

- Figure 1 comments:

- o Numbers do not add up – 11 are stated to be excluded but numbers add up to 10.

- o 201 enrolled and randomised but only 200 are shown to be assigned to groups. It’s shown that 1 withdrew consent but after randomisation, therefore they should still be included in the ‘assigned to’ row. Their withdrawal should be shown after this if they withdrew after randomisation. Please add an additional row to show the number of participants who received a vaccination.

- Figure 2 – I believe that this shows the maximum severity recorded for each participant for each reaction over the 14 days post-vaccination? Please make it clearer that this is the maximum severity.

- Figure 3 – please amend the error bars. These are the 95% CIs but are missing the lower bounds on many of the bars in panels b-d.
- Figure 4 comments:
 - o The top left plot in both Figure 4a and 4b are not on a log-scale. Please keep as consistent as possible and plot these on a log-scale, as with the rest of the anti-RBD IgG results.
 - o Significant markers in Figure 4b are confusing – were comparisons performed for each timepoint? If so, why do the significance bars only show for certain timepoints? If non-significant then please show the 'ns' bars as in panel a.
- Table 3 – the seroconversion rate for Ad5-nCoV day 90, the 95% CI does not cover the point estimate. Either the CI or the point estimate is incorrect and needs correcting.
- Table 5 – why are the authors looking at seroconversion rate rather than GMTs? Please consider looking at GMTs here instead.
- Table 5 – relationship between antibody levels and categorical BMI has been tested using a Chi-Squared test. Please perform a Cochran-Armitage test instead, to take the ordinal nature of the BMI variable into account.

Response to Reviewer 1's Comments

We feel great thanks for reviewer's professional review work on our article. As you are concerned, there are several problems that need to be addressed. According to reviewer's nice suggestions, we have made extensive corrections to our previous draft, and the detailed corrections are listed as follows. In response to your concerns about the exclusion on the basis of prior infection, we strongly agree with your points, but we have some relevant explanations. Our study was conducted from May to September 2022, at a special time when China had taken strict measures to prevent and control the epidemic, and the study regions had no large outbreaks and pandemics before. Additionally, our organization (Zhejiang CDC) had the recorded information of individuals who had been infected with SARS-CoV-2 based on the epidemic information system, so we choose exclusion criteria with a history of SARS-CoV-2 infection. This is unique to some clinical trials conducted in other countries.

Point 1: Introduction. The introduction is long and contains much generic information on the pandemic. The entire first paragraph, for example, could be omitted.

Response 1: Thanks for your great suggestion on improving the accessibility of our manuscript. We have deleted the first paragraph, and adjusted the description (Page 3, line 48).

“The breakthrough infection cases of coronavirus disease 2019 (COVID-19) are continuing in the real world...”

Point 2: On page 4, lines 77-79 be more specific about the heterologous boosters. From looking at the references, there were Ad5 and mRNA vaccines.

Response 2: We sincerely appreciate the valuable comments, we have added descriptions about heterologous boosters (Page 4, line 69-80).

“Immunity in those who had complete primary immunization was shown to be more efficiently restored by a heterologous booster than homologous boosters in clinical trials. In comparison to a third homologous dose of CoronaVac, recombinant adenoviral vectored vaccine, mRNA vaccine, or recombinant adenoviral-vectored ChAdOx1 nCoV-19 vaccine administration increased humoral and cellular immune responses^{16, 17, 18}. Additionally, boosting ChAdOx1-primed adults with SCB-2019 or mRNA vaccines induced higher levels of antibodies against a wild-type strain and SARS-CoV-2 variants than a homologous ChAdOx1 booster¹⁹. The adenovirus-vectored vaccine booster in individuals vaccinated with inactivated vaccines can be highly beneficial, Li et al. showed that administration of a heterologous boosting with AD5-nCOV following initial vaccination with CoronaVac was more immunogenic than homologous boosting²⁰.”

Point 3: Methods. Please provide additional detail on the vaccines, including the manufacturer. I am not familiar with the Covilo vaccine.

Response 3: Thanks for your great suggestion, we have added descriptions in methods (Page 6, line 125-132).

“Trial vaccines included CoronaVac (Vero Cell; Beijing Sinovac Research & Development, Beijing, China) and Covilo (Vero Cell; Beijing Institute of Biological Products Co., Beijing, China), are inactivated whole virion vaccine of wild-type SARS-CoV-2 with aluminum hydroxide as the adjuvant, which was administered intramuscularly at 0.5 mL per dose; Convidecia (CanSino Biologics, Tianjin, China), is a replication-defective Ad5-vectored vaccine expressing the full-length spike gene of wild-type SARS-CoV-2 (WuhanHu-1), supplied as a liquid formulation with a concentration of 5×10^{10} viral particles per 0.5 mL.”

Point 4: Page 5, line 100. How was a history of laboratory-confirmed SARS-CoV-2 determined?

Response 4: Thanks for your great suggestion. Our study was conducted from May to September 2022, when China had taken strict measures to prevent and control the epidemic. Our organization (Zhejiang CDC) had the recorded information of individuals who had been infected with SARS-CoV-2 based on the epidemic information system. Meanwhile, the study regions had no large outbreaks or epidemics during that period. We have added description in method section (Page 5, line 103).

“The exclusion criteria included a history of laboratory-confirmed SARS-CoV-2 infection (based on the epidemic information system in China)”

Point 5: Page 7, line 133. Why did the variety of the fourth dose of the inactivated vaccine be the same as that used for the third dose? Shouldn't these be interchangeable? Is this practical to implement in a public health program?

Response 5: Thanks for your great suggestion. We hold the opinion that the inactivated vaccine could be interchangeable, and it is permitted during vaccination practice in China. In the technical guidelines for SARS-CoV-2 vaccination (1st Edition) in China, it is recommended that the same vaccine product be used to complete the vaccination. In the event of special circumstances such as the vaccine being unavailable or the recipient being vaccinated in a different location, the same vaccine product from another manufacturer can be used to complete the vaccination (http://www.gov.cn/fuwu/2021-03/29/content_5596577.htm). The reason we designed the variety of the fourth dose same to the third dose was to reduce potential impact factors. We had enough clinical trial vaccines and the subjects were also willing to accept this. Additionally, we have compared the immunogenicity of sub-group of booster vaccination with Ad5-nCoV. The results showed that there was no statistic difference in antibody responses of Ad5-nCoV booster on those with a prime vaccinations of CoronaVac or Covilo (Figure 5).

Point 6: line 148. How were the anti-SARS-CoV antibody tests used, and what was the specific test? Were persons excluded based on this test?

Response 6: Thanks for your great suggestion. We have supplemented the assay of SARS-CoV-2-specific IgG (wild-type) and BA.4/5 pseudovirus-based neutralization antibody in the supplemental file (MATERIALS AND METHODS). We did not exclude persons based on this test. As we mentioned, the study region had no large outbreaks or epidemics during that period and before. We had accurate information on whether individuals were infected with SARS-CoV-2 or not, based on the epidemic information system in China. Additionally, some individuals who had a vaccination history may still have high antibody levels, therefore, it is difficult to distinguish antibody titres induced by infection or vaccination.

“SARS-CoV-2-specific IgG assay

The commercial anti-SARS-CoV-2 RBD-IgG ELISA detection kit (Vazyme Medical Technology, Nanjing, China) was employed to measure the levels of IgG against SARS-CoV-2 Receptor binding domain (RBD). Briefly, the serum specimen was diluted 3-fold with the sample diluent from 1:10 serially, in addition to the test wells, two negative control wells were included on each plate. After a 30-min incubation at 37°C away from direct light, each well was washed five times with diluted washing buffer, then filled with 100 µL of enzyme-labelled reagents and incubated again under the same conditions. After being washed as described above, each well was filled with 50 µL of chromogen solution A followed by 50 µL of chromogen solution B and then incubated at 37°C for 15 min. Finally, 50 µL of stop solution was added into each well, and the optical density (OD) of each well was measured via dual wavelength detection (at 450 nm/600–650 nm) on a spectrophotometer. The maximum dilution was the titer of the sample. The positive cutoff values for RBD-specific IgG antibodies were defined as titers of 1:90. The titer was converted to relative units per milliliter (RU/ml) with reference to the WHO international standard for anti-SARS-CoV-2 immunoglobulin

(NIBSC code 20/136). The positive RBD-specific IgG response was defined as a concentration ≥ 100 RU/ml.

Pseudovirus-based neutralization test

Serum samples were also quantified for their content of SARS-CoV-2-neutralizing antibodies to Omicron BA.4/5 using the pseudovirus-based virus neutralization test (a vesicular stomatitis virus pseudovirus system that expresses the spike glycoprotein). Briefly, serum samples and a positive or negative reference sample were each diluted 3 times with phosphate-buffered saline combined with 50 μ l of pseudovirus diluent per well in a 96-well plate. The mixed sample/pseudovirus was incubated at 37°C and 5% CO₂ for 1 h. A 2×10^5 /ml BHK-21-ACE2 cell suspension was added to each well of the plate containing the sample/pseudovirus mixture, then the plate was incubated in a 37°C and 5% CO₂ cell incubator for 48 h. Finally, the number of green-fluorescence-protein-positive cells per well was read with a porous plate imager (Tecan, Shanghai, SparkCyto). 50% neutralization titer (NT₅₀, the reciprocal of the dilution at 50% inhibition) calculated using the Reed-Muench method with a positive cutoff NAb titer $\geq 1:80$. Seroconversion was defined as at least a four-fold increase in antibody levels over the baseline values.”

Point 7: Statistical analysis, line 175. I assume you mean the sample size was 200, or the study was "designed to include" 200? Define the per-protocol set.

Response 7: Thanks for your great suggestion. We mean the sample size was 200. We have added the sample size calculation and defined per-protocol set in method section (Page 10, line 208-209, 212-216). “All participants who received the booster dose were included in the safety population (Safety set (SS)).” “The immunogenicity objectives are reported based on the per-protocol set (PPS), and missing data were not imputed. Participants experience no major protocol violation, comply with all inclusion criteria/exclusion criteria, complete the vaccination within the time window as required in the protocol and complete all blood samplings. ”

Point 8: Page 10, line 208. Please specify months.

Response 8: Thanks for your great suggestion. We have revised the sentence (Page 12, line 244). “At enrollment and before receiving the vaccine booster (day 0)”

Point 9: Page 11, lines 233-237. This is an interpretation of the results and should be moved to the discussion section.

Response 9: Thanks for your great suggestion. We have moved the description to the discussion section (Page 18, line 396-399).

“Similarly, our data indicated that adverse reactions resulting from heterologous boosting with Ad5-nCoV after three-dose priming with an inactivated SARS-CoV-2 vaccine are predictable and manageable, although they presented at a higher incidence than those induced by homologous boosting with an inactivated vaccine.”

Point 10: Page 13, lines 285 to end of paragraph. Likewise, this is a conclusion and should be in the discussion section.

Response 10: Thanks for your great suggestion. We have deleted the sentence and this has been described in the discussion section (Page 17, line 363-369; Page 18, line 380-384).

“Data in this study showed that the participants’ humoral responses were rapidly and robustly elevated by the Ad5-nCoV heterologous fourth-dose booster on day 14 after immunization, not only against WT SARS-CoV-2 but also against the Omicron BA.4/5 variant. Homologous boosting with the inactivated SARS-CoV-2 vaccine, however, elicited weaker antibody responses, especially in terms of neutralizing activity to Omicron BA.4/5.”

“Our data herein revealed that, in a scheme involving a fourth dose following three doses of inactivated vaccine, there was a greater decline in antibody responses against WT SARS-CoV-2 and Omicron BA.4/5 elicited by the inactivated vaccine on day 90 after the booster, whereas Ad5-nCoV maintained comparatively high levels.”

Point 11: page 18, I am confused by the mention of screening of histories. The serostatus of an individual and prior infection are important predictors of response to vaccine (those with prior infection generally having more robust responses). I can't follow how this was determined - through history and antibody testing? Or only history? And how were participants excluded?

Response 11: We agree with your point “The serostatus of an individual and prior infection are important predictors of response to the vaccine (those with prior infection generally having more robust responses)”. As this study was at a special time that no pandemic in China yet, the study regions had no large outbreaks, and our organization (Zhejiang CDC) had the recorded information of individuals who had been infected with SARS-CoV-2 based on the epidemic information system, so we choose exclusion criteria with history of SARS-CoV-2 infection. This is unique to some clinical trials conducted in other countries.

Point 12: Table 2 is redundant and not needed in my view.

Response 12: Thanks for your great suggestion. We have added p value in Table 2, and a mass of safety data showed in Table 2, so we hope to retain.

We would like to take this opportunity to thank you for all your time involved and this great opportunity for us to improve the manuscript. We look forward to hearing from you regarding our submission. We would be glad to respond to any further questions and comments that you may have.

Sincerely,

The Authors

Response to Reviewer 2 Comments

Thank you very much for your time involved in reviewing the manuscript and your very encouraging comments on the merits. We would like to take this opportunity to thank you for all your time involved and this great opportunity for us to improve the manuscript.

Sincerely,

The Authors

Response to Reviewer 3 Comments

According to the reviewers' comments, we have made extensive modifications to our manuscript and supplemented extra data to make our results convincing. Thank you again for your positive comments and valuable suggestions to improve the quality of our manuscript.

Point 1: Hypothesis testing/analysis comments:

- The manuscript states there was no sample size calculation. How was sample size determined if no formal sample size was performed? Was the sample size pre-determined? Can the authors please clarify this, as there are these inconsistencies between the manuscript and the protocol: both the protocol and the study SAP contain a power calculation – is this post-hoc, or based on pre-determined sample size?

Response 1: We were really sorry for our careless mistakes. The authors who wrote this section did not comprehend and carefully refer to the protocol and study SAP. The sample size was pre-determined before we start the clinical trial. We have added the sample size calculation to the statistical analysis section (Page 10, line 191-204).

“Sample size calculation was based on the assumption that heterologous booster immunization with Ad5-nCoV after three-dose priming with inactivated SARS-CoV-2 vaccine would elicit non-inferior and superior concentrations of neutralizing antibodies to the homologous booster dose with inactivated SARS-CoV-2 vaccine. We assumed that the one-sided α of the inspection level = 0.025, the ratio of Ad5-nCoV group to inactivated SARS-CoV-2 vaccine group was 1: 1, the non-inferiority limit of GMT ratio was 0.67 (after 10 is the bottom log = -0.174), and the actual GMT ratio of the two groups was 2, the study needed to recruit 80 participants in each group to achieve 99.86% power. When the sample size of each group was 80, the superiority of GMT level of Ad5-nCoV group was greater than that of the inactivated SARS-CoV-2 vaccine group, and the estimated power of the test is 88.16%, which meets the test requirements. Thus, we decided on a total sample size of 200 with 100 for each group after adjusting for an attrition rate of 20% due to loss to follow-up. Power Analysis and Sample Size software (LLC, USA version 11.0.7) was used.”

Point 2: The protocol and SAP both show that the main hypothesis under evaluation was non-inferiority of Ad5-nCoV to inactivated vaccine control, based on geometric mean ratio (GMR) of the GMT for the two arms (for the anti-RBD IgG results, judging by the values in the SAP section 2.3). This manuscript contains no GMR results and therefore doesn't address the hypotheses as stated in the protocol. The anti-RBD day 14 GMR with CI needs to be provided, as planned in the SAP. The non-inferiority comparison needs to be made to then justify the superiority comparison. I would suggest that the GMR should be adjusted for baseline (pre-booster) levels of anti-RBD. Otherwise, a reason for not performing the primary hypothesis needs to be stated in the text and justified.

Response 2: Thanks for your great suggestion. Based on your comments, we have added the results of anti-RBD GMR with CI on day 14 (Page 14, line 294-297).

“Similarly, boosting with Ad5-nCoV significantly increased RBD-specific IgG antibody responses with a GMFIs of 34.8 (95% CI 26.5–45.7) than inactivated vaccine with a GMFIs of 3.7 (95% CI 3.0–4.6) on day 14 ($P < 0.0001$) (Fig. 3d).”

Point 3: There is no information on the numbers of participants who received CoronaVac or Covilo as their third dose (and therefore as their trial dose) – please include this information in both the study flow diagram (Figure 1) and baseline characteristics table (Table 1). I would suggest the authors to provide a supplementary table of control group GMTs split by CoronaVac and Covilo recipients, so readers can see similar results between these sub-groups (justifying pooling both vaccines into one ‘inactivated vaccine’ control group).

Response 3: Thanks for your great suggestion. We have added the information on the number of participants who received CoronaVac or Covilo to Figure 1 and Table 1. “62 participants received CoronaVac and 38 participants received Covilo.” Additionally, we have compared GMTs of CoronaVac and Covilo recipients of control group in supplementary table 1 and 2, and Figure 1. The results showed CoronaVac induced higher antibody levels than Covilo. It maybe due to the different age composition and lower antibody levels of base line. However, the seroconversion and GMFI had no statistic difference. So we think both vaccines can be divided into one ‘inactivated vaccine’ control group. Supplementary table 1 and 2, and Figure 1

Supplementary Table 1 Baseline characteristics of enrolled participants.

Variable ^a	CoronaVac (n=62)	Covilo (n=38)	p
Sex			0.09
Male	30(48.4)	25(65.8)	
Female	32(51.6)	13(34.2)	
Age			<0.001
18-59 years	46(74.2)	13(34.2)	
≥60 years	16(25.8)	25(65.8)	
Median age (IQR), years	44.0(36.8,60.0)	63.0(42.8,66.5)	
Body mass index (kg/m²)			0.426
≤18.4	0(0)	1(2.6)	
18.5-24.9	42(67.8)	23(60.5)	
25.0-29.9	17(27.4)	13(34.3)	
≥30.0	3(4.8)	1(2.6)	
Time interval since the last priming dose of inactivated vaccine, months			<0.001
Median (IQR)	6.8(6.5,7.0)	6.4(6.2,6.7)	
Underlying chronic diseases^b			0.294
Yes	19(30.6)	8(21.1)	
No	43(69.4)	30(78.9)	

^aData are the number of participants (%), or median (IQR).

^bUnderlying chronic diseases included cardiovascular and cerebrovascular diseases, hypertension, chronic obstructive pulmonary disease, etc.

Point 4: I would suggest for the authors to perform an exploratory immunogenicity sub-group analysis by prime vaccination schedules (CoronaVac and Covilo), to observe if there are differences in the effect of Ad5-nCoV booster on those with a prime series of CoronaVac vs Covilo, with adjustments made for differences in participant populations. Potentially obvious limitations are sample size, but it would still be good to perform this exploratory analysis to observe if there are any detectable differences.

Response 4: Thank you for your suggestions. We have added immunogenicity sub-group analysis (Page 15, 328-331)

“Additionally, we explored the immunogenicity of sub-group of booster vaccination with Ad5-nCoV. The results showed that there was no statistic difference in antibody responses of Ad5-nCoV booster on those with a prime vaccinations of CoronaVac or Covilo (P > 0.05) (Figure 5).” Figure 5 and supplementary table 3 and 4.

Point 5: Can the authors please justify their choice in using Kruskal-Wallis tests? These non-parametric tests are performed on log-transformed data, which should be normally distributed. Please consider using a parametric test to analyse this data.

Response 5: We sincerely thank the reviewer for careful reading. We feel sorry for our carelessness. We used the One-way ANOVA with Tukey's multiple comparisons test in fact. We have revised the descriptions in method and Figure 1 legend (Page 10, line 220-222).

“We used Student’s t-test and One-way ANOVA with Tukey's multiple comparisons test comparisons test to analyze the log-transformed antibody titers and categorical data.”

Point 6: Can the authors please comment on SAEs – this is a secondary endpoint but has not been mentioned in the results section. If no SAEs occurred then please state this. Could the authors please also justify why SAEs were only collected for the first 28 days post-vaccination?

Response 6: We collected the SAEs throughout the study (finished the last follow-up visits), and the unsolicited adverse events were collected within 28 days post vaccination (Page 9, line 184-187). “Safety secondary outcomes included information on unsolicited adverse events for 28 days after immunization and serious adverse events recorded throughout the study.”

And we had mentioned the results of SAEs in result section (Page 13, line 267-269). “No thromboses, vaccine-related anaphylaxis, or other serious adverse events were documented during follow-up visits.” Thank you for your suggestions. We have described it more clear (Page 9, line 184-187). “Safety secondary outcomes included information on unsolicited adverse events for 28 days after immunization and serious adverse events recorded throughout the whole follow-up visits.”

Point 7: Population comments:

- 200 participants received a vaccination on the trial, but the safety population only includes 199 participants. Line 176-177 states “all participants who received the booster dose were included in the safety population”. Why is this one participant missing from the safety analysis? Please explain this in the text and in Figure 1. If this is the participant that discontinued then did they discontinue from all safety follow-ups immediately after vaccination? (If not then there should be data, and if so then isn't this a withdrawal rather than a discontinuation?)

Response 7: Thanks for your great comment. Although this participant discontinue from all safety follow-ups after vaccination, but as a safety set (SS), we should still calculate it in the denominator (200). It was our miscalculation and we have revised in Figure 1, 2 and Table 2.

Table 2 Solicited and unsolicited adverse reactions^a.

Variable	Inactivated vaccine (n=100)	Ad5-nCoV (n=100)	p
All solicited adverse reactions within 0–14 days^b			
Total	25(25.0)	55(55.0)	<0.001
Grade1	22(22.0)	32(32.0)	0.111
Grade2	3(3.0)	13(13.0)	0.009
Grade3	0(0.0)	10(10.0)	0.001
Injection site adverse reactions within 0–14 days			
Total	18(18.0)	44(44.0)	<0.001
Pain	16(16.0)	39(39.0)	<0.001
Induration	3(3.0)	13(13.0)	0.121
Redness	3(3.0)	8(8.0)	0.121
Swelling	3(3.0)	11(11.0)	0.009
Grade 3	0(0.0)	2(2.0)	0.497
Rash	1(1.0)	0(0.0)	1.000
Itch	2(2.0)	11(11.0)	0.010
Systemic adverse reactions within 0–14 days^c			
Total	12(12.0)	35(35.0)	<0.001
Nausea	0(0.0)	1(1.0)	1.000
Fever	2(0.0)	20(20.0)	<0.001
Grade 3	0(0.0)	7(7.0)	0.014
Diarrhea	3(3.0)	4(4.0)	1.000
Arthralgia	0(0.0)	12(12.0)	<0.001
Cough	0(0.0)	5(5.0)	0.059
Runny nose	3(3.0)	4(4.0)	1.000

Point 8: Was there an upper limit on recruitment age (80 years)? The inclusion criteria states ≥ 18 years but further details in the SAP and protocol show an upper limit of 80 years. If so, please make this clear in the manuscript.

Response 8: We are sorry for our carelessness. We have revised it in the manuscript.

Point 9: Can the authors please explain the justification for their 2:1 ratio for 18-59 and ≥ 60 years age groups. On what basis was this chosen, particularly given that the authors state a fourth dose is recommended by the WHO only to those ≥ 60 years?

Response 9: Thank you for your suggestions. We chose the 2:1 ratio for 18-59 and ≥ 60 years age groups was based on percentage of age composition of the population. The purpose of the study is focus on the whole adult population, not only include WHO recommended old people (≥ 60 years and), but young adults who may engage in medical and health care and disease prevention and control, which are high-risk population infected with SARS-CoV-2.

Point 10: The immunogenicity analysis is stated to have been performed on the per-protocol dataset. The protocol and SAP contain definitions for this, but they do not state if blood draws needed to be within the pre-specified windows (as in the protocol, section 8.8.1). Please include a clear definition of the per-protocol dataset in the manuscript.

Response 9: Thank you for your suggestions. We have added the description about definition of the per-protocol dataset(Page,10 line 212-216).

“The immunogenicity objectives are reported based on the per-protocol set (PPS), and missing data were not imputed. Participants experience no major protocol violation, comply with all inclusion criteria/exclusion criteria, complete the vaccination within the time window as required in the protocol and complete all blood samplings.”

Point 11: Randomisation comments:

- The protocol states that block randomisation was used. Please include details of this in the manuscript, with details of block sizes used.

Response 11: Thank you for your suggestions. We have added the descriptions about block randomisation (Page7, line 133-139).

“The block randomization method was used with block sizes of 2, and stratified by age group. Eligible participants in each cohort were randomly assigned in a 1:1 ratio to receive Ad5-nCoV or inactivated vaccine. Randomization was done using a sealed enveloped system integrated with the electronic case report forms in the Open Clinica platform. A random number was assigned to each participant by allocation of the next available randomized entry in the randomization list (generated by an independent statistician using SAS (version 9.4)) that was established before the start of the study. ”

Point 12: Randomisation was stratified by age group – please include this detail in the ‘Randomization and Masking’ section of the manuscript.

Response 12: Thank you for your suggestions. We have added the descriptions about Randomization and Masking (Page7, line 133-139).

“The block randomization method was used with block sizes of 2, and stratified by age group. Eligible participants in each cohort were randomly assigned in a 1:1 ratio to receive Ad5-nCoV or inactivated vaccine. Randomization was done using a sealed enveloped system integrated with the electronic case report forms in the Open Clinica platform. A random number was assigned to each participant by allocation of the next available randomized entry in the randomization list (generated by an independent statistician using SAS (version 9.4)) that was established before the start of the study. ”

Point 13: Was randomisation stratified by sex as well as age? If so, please also mention this in the randomisation section.

Response 13: Thank you for your suggestions. Randomisation was stratified by age group. Refer to the preceding

Point 14: Discussion comments:

- Can the authors please include in their discussion how the results of this trial will translate to the current population, given that the study is on a seronegative population but this no longer reflects the state of the current population.

Response 14: Thank you for your suggestions. We have add the discussion (Page10, line 430-434).

“Fourth, the study was conducted on a seronegative population but no longer reflected the state of the current population. However, as a strategy of COVID-19 vaccination, our results suggested heterologous booster is more effective than homologous booster for enhancing antibody levels, which is also applicative for population who have been been infected.”

Point 15: The authors show a difference in anti-RBD IgG between the 2 age groups in the inactivated vaccine group, but not in participants receiving Ad5-nCoV. They commented that the same results were not seen when observing neutralising antibody. This is an interesting finding – please can the authors further comment on this in the discussion.

Response 15: We feel sorry that we had no comment about the results were different between anti-RBD IgG and neutralising antibody. There may be a misunderstanding here, our results suggested that the influence of age on antibody levels was less apparent in the Ad5-nCoV group than the inactivated vaccine group, especially in the early days following booster vaccination by both anti-RBD IgG and neutralising antibody.

Point 16: Can the authors please include in the discussion how their findings might translate to protection against infection. This is mentioned as a limitation but it needs further discussion. Many published studies show a difference in vaccines that doesn't translate to a difference in protection against hospitalisations or severe disease.

Response 16: Thank you for your suggestions. We agree with you completely. The immunogenicity is substitution to evaluate the immune effect of vaccines, which is difficult to translate to protection against infection. The protection of vaccine is influenced by many factors. The protection or duration of protection of vaccines should evaluate in real-world observational studies with large population. We have added the limitation in discussion section (Page20, line 434-437).

“Finally, the study was an immunogenicity evaluation, which could not translate to a difference in protection against hospitalizations or severe disease. There has been no direct correlation between immunization and protection or duration of protection²¹”

Point 17: Related to the point above, on line 387 please remove the statement ‘more efficient’. This implies that Ad5-nCoV has higher vaccine efficacy compared to inactivated vaccine as fourth dose, but there is no direct evidence for this. Although immunogenicity is clearly higher in the Ad5-nCoV arm, this doesn't necessarily translate to more protection against severe disease/hospitalisation (e.g. seen in the following article, when comparing vaccine efficacy of BNT162b2 and CoronaVac in the population: <https://www.sciencedirect.com/science/article/pii/S1473309922003450?via=ihub>)

Response 17: Thank you for your suggestions. We have deleted the word “more efficient”.

Point 18: Please can the authors add to their statement in their limitations section of not looking at cellular immune data – although cellular outcomes weren't looked at, please comment on what based on the current literature on cellular response after Ad5-nCoV vaccination.

Response 18: Thank you for your suggestions. We have added the statement as you comment (Page20, line 428-430).

“it would be expected that heterologous Ad5-nCoV booster enhanced cellular immune response than homologous booster of inactivated vaccine²⁰”

Point 19: Line 324-325 – the authors state the “Ad5-nCoV booster also dramatically increased the decay time compared to the homologous inactivated vaccine”, however the rate of decay isn’t a result looked at in this manuscript and there are only results up to 90 days following vaccination. Do the authors mean that the antibody levels at day 90 following booster vaccination were significantly higher in participants receiving Ad5-nCoV than those receiving inactivated vaccine? Please can the authors change their wording here to help clarify.

Response 19: Thank you for your suggestions. The sentence we described not clearly, and we have revised as your comments (Page17, line 372-373).

“The antibody levels following homologous booster of inactivated vaccine in participants decayed faster than heterologous Ad5-nCoV booster.”

Point 20: Lines 338-341 – it’s stated that “our results showed the fourth-dose Ad5-nCoV booster further elevated the peak value following three doses of the inactivated SARS-CoV-2 vaccine”. Please change “showed” to “suggested” – as the authors already state they have no data on peak value following three doses to show this.

Response 20: Thank you for your suggestions. We have revised the word.

Point 21: Line 386 – please change “safe” to ‘tolerable’, for example. “Safe” cannot be stated in this small sample, given that there is insufficient power to detect rare safety events (and SAEs were only collected for 28 days following trial vaccination).

Response 21: Thank you for your suggestions. We have revised the word.

Point 22: Other comments:

- Can the authors please clarify what is meant by “duration of antibody levels” (lines 86, 267)? Is this duration above a certain threshold, or above pre-vaccination levels?

Response 22: The duration included both “above a certain threshold”, and “above pre-vaccination levels”. The evaluation indicators contain seropositivity means “above a certain threshold”. GMFI means geometric mean fold increase compared pre-vaccination levels, and seroconversion rate is defined as percentage value of at least a four-fold increase in antibody levels over the baseline values mean above pre-vaccination levels.

Point 23: Line 36 – please change statement “maintained antibody levels for longer” to “maintained higher antibody levels at day 90”. Or instead please clarify that they are talking about levels above the seropositive threshold.

Response 23: Thank you for your suggestions. We have revised the sentence (Page2, line 36-37).

“The Ad5-nCoV booster-maintained a high antibody levels on day 90, with seroconversion of 71.4%,”

Point 24: Line 102 – can the authors please clarify if “receipt of any vaccine within 30 days before or after each study dose” should refer to each dose prior to the study, not the on-study dose? Otherwise this criteria overlaps with the next (lines 103-104) and is confusing.

Response 24: We feel sorry for our carelessness. This criteria is overlaps in Line 102 and lines 103-104. We have revised and deleted the sentence “receipt of any vaccine within 30 days before or after each study dose”.

Point 25: Did any of the assays have lower or upper limits of detection (definitely appears that anti-RBD IgG assay had lower limit of detection)? If so, please include in the text how these were dealt with in the immunogenicity analysis (i.e. were values below or above limits of detection imputed?).

Response 25: Thank you for your suggestions. There was a lower limit of detection in our assays. We feel sorry not mention it. The values below limit were imputed and defined the value of detection limit. We have revised the descriptions in method section and figures (Page 8-9, line 166-177).

“The lower limit of detection was 1:10, values below limit were imputed and defined 1:10.”

“The lower limit of detection was 10 RU/ml, values below limit were imputed and defined 10 RU/ml.”

Point 26: Please mention the different inactivated vaccines in lines 29 (abstract) and 83-84 (introduction) – it is not mentioned until line 123 currently.

Response 26: Thank you for your suggestions. We have revised the sentences (Page 2, line 30-32; Page 4, line 83-87).

“then administered intramuscular Ad5-nCoV or different inactivated SARS-CoV-2 vaccine (CoronaVac or Covilo) respectively.”

“Here, we present the safety and immunogenicity results following heterologous booster immunization with Ad5-nCoV or homologous boosters with a different inactivated SARS-CoV-2 vaccine (CoronaVac or Covilo) after three-dose priming with the inactivated vaccine in healthy participants aged ≥ 18 years in a randomized, double-blind, parallel-controlled phase 4 trial.”

Point 27: Please can the authors change their use of the word “robust” in this manuscript (lines 40, 316, 388)? The use of “robust” is subjective and unclear.

Response 27: Thank you for your suggestions. We have corrected the “robust” to “strong” (Page 2, line 41; Page 17, line 364; Page 21, line 444)

Point 28: I would suggest that seroconversion is defined in line 165 (it is not defined until line 241).

Response 28: Thank you for your suggestions. There was a sentence we described as a definition of Seroconversion before “line 165” (Page 9, line 177-178).

“Seroconversion was defined as at least a four-fold increase in antibody levels over the baseline values.”

Point 29: What tests were used to produce the p-values stated for the safety analysis (lines 215, 217, 219)? Please include this in the Statistical Analysis section after line 178 (all following information regarding statistical tests in this section relates only to the immunogenicity analysis).

Response 29: Thank you for your suggestions. We have revised in the Statistical Analysis section (Page 10, line 211-212)

“The analysis was based on the intention-to-treat cohort, and calculated with Chi-Squared test or Fisher’s exact test.”

Point 30: Line 196 is the only time it’s mentioned that this is a phase 4 trial. Suggest including this earlier in the manuscript, e.g. in line 91.

Response 30: Thank you for your suggestions. We have revised it (Page 4, line 83-87).

“Here, we present the safety and immunogenicity results following heterologous booster immunization with Ad5-nCoV after three-dose priming with the inactivated vaccine in healthy participants aged ≥ 18 years in a randomized, double-blind, parallel-controlled phase 4 trial.”

Point 31: The text should state that no grade 4 solicited events were reported in the trial. Please also note this in Table 2, at least in the footnote.

Response 31: Thank you for your suggestions. We have added the descriptions in result section (Page 13, line 266-269) and Supplementary Table 1.

“There were no grade 4 solicited events were reported in the trial, and no thromboses, vaccine-related anaphylaxis, or other serious adverse events were documented during follow-up visits.”

Point 32: Many additional solicited adverse reactions were collected but not reported in the manuscript. Although this is likely due to these reactions not occurring, it is still important to relay this information. Please add to the footnote on Table 2 stating that no events were reported for cellulitis, vomiting, chest pain, etc. (stating all the solicited adverse reactions collected, for which no events were reported). Please add a similar footnote to Figure 2.

Response 32: Thank you for your suggestions. We have revised as your comments.

Table 2:^{ab}There was no grade 4 solicited events were reported in the trial. cno events were reported for cellulitis, vomiting, chest pain, etc.”

Figure 2:

* No events were reported for cellulitis, vomiting, chest pain, etc.

Point 33: Differences in sex, age, chronic diseases were explored in the relationship between vaccine and safety events, but no adjustment for multiple testing was used. Could the authors please justify this? Please consider adjusting for multiple testing. Same comment applies for looking at factors influencing antibody levels. (Table 4 and table 5.)

Response 32: Thank you for your suggestions. We have analyzed the influencing factors of solicited adverse reactions with logistic regression model of multivariate analysis in Table 4. Age of 60-80 years is the main protective factor of adverse reaction.

Table 4: Analysis of influencing factors of solicited adverse reactions.

	Inactivated vaccine (n=99*)			Ad5-nCoV (n=100)				
	Adverse reactions rates %	Chi-Squared test	Logistic test		Adverse reactions rates %	Chi-Squared test	Logistic test	
		p	OR	95%CI		p	OR	95%CI
Sex		0.023				0.034		
Male	9(16.4)		ref		25(45.5)		ref	
Female	16(36.4)		1.74	0.61-4.90	30(66.7)		2.20	0.91-5.32
Age		<0.001				<0.001		
18-59 years	23(39.0)		ref		42(70.0)		ref	
60-80 years	2(4.9)		0.12	0.03-0.56	13(32.5)		0.325	0.11-0.94
Body mass index (kg/m2)		0.158	-	-		0.157	-	-
≤18.4	0(0.0)				4(100.0)			
18.5-24.9	20(30.8)				31(57.4)			
25.0-29.9	5(16.7)				15(44.1)			
≥30.0	0(0.0)				5(62.5)			
Underlying chronic diseases		0.016				0.001		
Yes	2(7.7)		ref		10(31.3)		ref	
No	23(31.5)		0.39	0.08-2.02	45(66.2)		0.48	0.16-1.48
Third dose of Inactivated vaccine		0.262	-	-		0.050	-	-
CoronaVac	18(29.0)				42(61.8)			
Covilo	7(18.9)				13(40.6)			

*Not calculate one participant who did not provide information of solicited adverse reactions within 0-14 days. Logistic regression model (Enter method) was used to conduct multivariate analysis to further explore the influencing factors of solicited adverse reactions rates.

The inclusion criterion $\alpha=0.05$, and the exclusion criterion $\alpha=0.10$. Calculate the OR and its 95%CI for "Yes" versus "no". Test level $\alpha=0.05$.

Point 34: Line 238 – please remove the word ‘Approximately’.

Response 34: We have removed the word ‘Approximately’.

Point 35: Please provide units for GMTs in both the text and for Table 3.

Response 35: Thank you for your suggestions. There is no units for GMTs. GMT, geometric mean titer, was calculated as the antilogarithm of the mean of the log₁₀ transformed SARS-CoV-2 RBD-specific IgG antibodies and Omicron BA.4/5 pseudovirus-neutralizing antibodies, showing results of such as 1:80, 1:200, etc.

Point 36: Suggest that lines 354-356 should be made clearer. Currently it reads as though a combination of these 3 factors are associated with lower frequency of adverse reactions, as opposed to each of these factors individually.

Response 36: Thank you for your suggestions. We have revised as your comments(Page,19,line 404-406) “Interestingly, participants who had a combination of male, aged 60-80 years old, and had chronic diseases had a lower frequency of local and systemic adverse reactions.”

Point 37: • Figure 1 comments:

o Numbers do not add up – 11 are stated to be excluded but numbers add up to 10.

o 201 enrolled and randomised but only 200 are shown to be assigned to groups. It’s shown that 1 withdrew consent but after randomisation, therefore they should still be included in the ‘assigned to’ row. Their withdrawal should be shown after this if they withdrew after randomisation. Please add an additional row to show the number of participants who received a vaccination.

Response 37: Thanks for your careful checks. We are sorry for our carelessness. We have revised the Figure 1.

Point 38: Figure 2 – I believe that this shows the maximum severity recorded for each participant for each reaction over the 14 days post-vaccination? Please make it clearer that this is the maximum severity.

Response 38: Thank you for your suggestions. We have revised the descriptions in Figure 2 legend.

“Fig. 2 Solicited adverse reactions over the 14 days post-vaccination. Analysis was based on the safety cohort, which included all randomly assigned participants who received the booster vaccination. The maximum severity of solicited adverse reactions recorded for each participant for each reaction. C = Inactivated SARS-CoV-2 vaccine group. T = Ad5-nCoV group.”

Point 39: • Figure 3 – please amend the error bars. These are the 95% CIs but are missing the lower bounds on many of the bars in panels b-d.

Response 39: We feel sorry for our carelessness. Based on your comments, we have made the corrections in Figure 3.

Point 40: • Figure 4 comments:

o The top left plot I both Figure 4a and 4b are not on a log-scale. Please keep as consistent as possible and plot these on a log-scale, as with the rest of the anti-RBD IgG results.

o Significant markers in Figure 4b are confusing – were comparisons performed for each timepoint? If so, why do the significance bars only show for certain timepoints? If non-significant then please show the ‘ns’ bars as in panel a.

Response 39: Thank you for your suggestions. We have revised in Figure4 as you comment.

Point 41: • Table 3 – the seroconversion rate for Ad5-nCoV day 90, the 95% CI does not cover the point estimate. Either the CI or the point estimate is incorrect and needs correcting.

Response 41: Thanks for your careful checks. We are sorry for our carelessness. Based on your comments, we have made the corrections in Table3.

Table 3 SARS-CoV-2 RBD-specific IgG antibodies and pseudovirus-neutralizing antibodies to Omicron BA.4/5 before and after booster vaccination

Variable	Inactivated vaccine				Ad5-nCoV			
	Day 0 (n=99 ^a)	Day 14 (n=99 ^a)	Day 28 (n=98 ^a)	Day 90 (n=97 ^a)	Day 0 (n=100)	Day 14 (n=100)	Day 28 (n=99 ^a)	Day 90 (n=98 ^a)
Anti-RBD-IgG								
GMT	63.1 (48.1-82.7)	263.3 (191.0-292.4)	170.5 (138.6-209.8)	106.7 (83.7-135.9)	64.7 (50.7-82.7)	2250 (1806.0-2803.0)	1594 (1284.0-1979.0)	993.2 (786.4-1254.0)
GMC (RU/mL)	119.1 (69.9-168.2)	333.3 (247.8-418.8)	234.4 (182.6-286.2)	152.7 (104.7-200.7)	105.2 (77.5-132.8)	2924.0 (2305.0-3542.0)	2142.0 (1649.0-2635.0)	1275.0 (868.3-1681.0)
Seropositive rate (%)	52.5 (42.2-62.7)	93.9 (87.3-97.7)	90.8 (83.2-95.7)	76.3 (66.6-84.3)	55.0 (44.7-65.0)	100.0 (96.4-100.0)	100.0 (96.3-100.0)	100.0 (96.3-100.0)
Seroconversion rate (%)	NA	37.4 (27.9-46.7)	27.6 (19.0-37.5)	5.2 (1.7-11.6)	NA	91.0 (83.6-95.8)	88.9 (81.0-94.3)	71.4 (61.4-80.1)
GMFI	NA	3.7 (3.0-4.6)	2.7 (2.2-3.4)	1.7 (1.4-2.1)	NA	34.8 (26.5-45.7)	25.0 (19.0-32.8)	9.1 (7.0-11.8)
Neutralizing antibodies to Pseudovirus (BA.4/5)								
GMT	34.1 (31.0-37.4)	65.5 (58.1-76.0)	51.7 (45.1-59.2)	31.9 (27.7-36.8)	30.0 (27.2-33.1)	228.9 (187.4-279.5)	162.1 (131.7-199.6)	99.8 (82.8-120.2)
Seropositive rate (%)	4.0 (1.1-10.0)	34.3 (25.1-44.6)	26.5 (18.1-36.4)	8.2 (3.6-15.6)	1.0 (0.02-5.4)	79.0 (69.7-86.5)	74.7 (65.0-82.9)	61.2 (50.8-70.9)
Seroconversion rate (%)	NA	17.2 (10.3-26.1)	8.2 (3.6-15.5)	0.0 (0.0-3.7)	NA	74.0 (64.3-82.3)	58.6 (48.2-63.4)	38.8 (29.1-49.2)
GMFI	NA	2.0 (1.7-2.3)	1.1 (0.9-1.3)	1.0 (0.8-1.1)	NA	7.6 (6.3-9.3)	5.4 (4.4-6.6)	3.3 (2.7-4.0)

Point 42: Table 5 – why are the authors looking at seroconversion rate rather than GMTs? Please consider looking at GMTs here instead.

Response 42: Thank you for your suggestions. The main consideration is the chi-square test to analyze the influencing factors of antibody levels. Based on your suggestion, we have also added the statistic analysis of GMTs in Table 5.

Table 5 Analysis of influencing factors of antibody levels (Omicron BA.4/5).

Variable	Inactivated vaccine (n=99)				Ad5-nCoV (n=100)						
	Seropositivity rates %	P	Seroconversion rates %	P	GMT (95%CI)	P	Seropositivity rates %	P	GMT (95%CI)	P	
Sex		0.219		0.851		0.127		0.786		0.780	0.557
Male	16(29.1)		9(16.4)		60.6 (51.5-71.3)		44(80.0)		40(72.7)		216.8 (168.1-279.7)
Female	18(40.9)		8(17.8)		74.7 (59.6-93.6)		35(77.8)		34(75.6)		244.5 (176.4-338.9)
Age		0.013		0.108		0.045		0.193		0.193	0.670
18-59 years	26(44.1)		4(9.8)		74.3 (61.7-89.6)		50(83.3)		45(75.0)		265.7 (204.1-345.9)
60-80 years	8(20.0)		13(22.0)		56.4 (47.0-67.7)		29(72.5)		29(72.5)		183.0 (134.9-248.2)
Body mass index (kg/m2)		0.090		0.535		0.084		0.157		0.421	0.218
≤18.4	0(0.0)		0(0.0)		48.0		4(100.0)		4(100.0)		395.0 (139.3-1120)
18.5-24.9	25(39.1)		12(18.5)		73.3 (61.9-86.8)		40(74.1)		37(68.5)		230.7 (174.5-305.1)
25.0-29.9	9(30.0)		5(16.7)		59.9 (47.0-76.3)		29(85.3)		26(76.5)		188.3 (140.7-252.1)
≥30.0	0(0.0)		0(0.0)		33.3 (22.8-48.5)		6(75.0)		7(87.5)		378.3 (107.5-1331)
Underlying chronic diseases		0.159		0.807		0.020		0.230		0.740	0.013
Yes	6(23.1)		5(18.5)		51.1 (38.8-67.2)		23(71.9)		23(71.9)		159.3 (115.2-220.2)
No	28(38.4)		12(16.4)		73.0 (62.8-85.0)		56(82.4)		51(75.0)		271.5 (212.3-347.1)
Adverse reactions		0.008		0.877		0.060		0.001		0.004	0.001
Yes	14(56.0)		4(16.0)		82.7 (63.5-107.9)		49(89.1)		47(85.5)		304.9 (237.9-390.7)
No	20(27.0)		13(17.3)		61.7 (52.9-72.1)		30(66.7)		27(60.0)		161.2 (119.0-218.3)
Third dose of Inactivated vaccine		0.757		0.136		0.567		0.501		0.519	0.731

Point 43: Table 5 – relationship between antibody levels and categorical BMI has been tested using a Chi-Squared test. Please perform a Cochran-Armitage test instead, to take the ordinal nature of the BMI variable into account.

Response 43: Thank you for your great suggestions. We have analyzed with Cochran-Armitage test again and revised it in Table 5 as your comment.

We tried our best to improve the manuscript and made some changes marked in red in revised paper which will not influence the content and framework of the paper. We appreciate for Reviewers' warm work earnestly, and hope the correction will meet with approval. Once again, thank you very much for your comments and suggestions. We would like to take this opportunity to thank you for all your time involved and this great opportunity for us to improve the manuscript.

Sincerely,

The Authors

Reviewers' Comments:

Reviewer #1:

Remarks to the Author:

Given the large number of people in the world who have received inactivated Covid-19 vaccines, and the limited experience on boosting these vaccines, this study has important policy implications. The revised version of the manuscript is improved, and most concerns have been adequately addressed. A few remaining edits are suggested below.

Introduction. What is SCB-2019 vaccine? A description of the type of vaccine would be helpful here. Statistical analysis. There is a typo after P value on page 11. "less than 0.05c".

Results.

Were all 211 volunteers recruited on May 14, or was this the day recruitment began?

Since the study was designed with two age groups, please provide the numbers in each age group and median age in each age group (not combined) in this first paragraph. Likewise, provide safety results (at least in a general statement) by age group.

Information on immunogenicity is provided by age group. It would be particularly interesting to know if safety differed by age group for the adenovirus boost, since immunogenicity did not.

Discussion. It would be worth commenting on the age differences in the discussion, since many countries are recommending additional boosters for older but not younger individuals.

Line 436 – "cloud" should be "could".

Line 445 – What is "cc"?

Given the current state of the field, it would be useful for the authors to comment on the benefit of updating vaccines to contain more recent Covid-19 variants. Is that what they mean by "next generation" vaccines?

Reviewer #3:

Remarks to the Author:

I'd like to thank the authors for addressing and responding to my comments and for the great deal of work they have put into their manuscript. I have only 2 comments and otherwise think that this is an important and valuable manuscript:

- Lines 278-297 – the addition of the GMFIs improves this section, however the GMT ratio (Geometric Mean Ratio [GMR]) between the arms (Ad5-nCoV/inactivated vaccine) at day 14 should be reported. The SAP states that the research hypothesis 1 is "the antibody level of Ad5-nCoV group 14 days after booster immunization is not inferior to that of ICV booster immunization group"; to answer the main hypothesis of the trial, the GMR of Ad5-nCoV/ICV at day 14 is what should be presented along with the 95% confidence interval. The lower bound of the confidence interval can then be assessed to determine if it meets the criteria to claim non-inferiority (>0.67).

- Lines 404-406 – my original comment was perhaps unclear, so my apologies to the authors. I believe the authors wish to say that each of the factors (male, aged 60-80, chronic disease) *individually* are associated with lower frequency of adverse reactions, not necessarily in combination. If so, this sentence needs to be reworded.

Response to Reviewer 1's Comments

According to the reviewers' comments, we have made extensive modifications to our manuscript to make our results convincing. Thank you again for your positive comments and valuable suggestions to improve the quality of our manuscript.

Point 1: Introduction. What is SCB-2019 vaccine? A description of the type of vaccine would be helpful here.

Response 1: Thanks for your great suggestion. We have added the description of the type of vaccine about SCB-2019 (Page 4, line 74-75).

“Additionally, boosting ChAdOx1-primed adults with SCB-2019 (a protein subunit vaccine, S-Trimer)...”

Point 2: Statistical analysis. There is a typo after P value on page 11. “less than 0.05c”.

Response 2: We sincerely thank the reviewer for careful reading. We feel sorry for our carelessness and have corrected it (Page 11, line 227).

“P-values of less than 0.05 were considered statistically significant...”

Point 3: Were all 211 volunteers recruited on May 14, or was this the day recruitment began?

Response 3: May 14, 2022 was the day recruitment began, and we finished the recruitment on May 27. We did not describe clearly. Thanks for your great suggestion, we have revised. (Page 11, line 234).

“Between May 14 and 27, 2022, 211 volunteers aged 18-80 years who had received three doses of inactivated vaccine (CoronaVac or Covilo) \geq 6-month earlier were recruited and screened for eligibility for this phase 4 trial.”

Point 4: Since the study was designed with two age groups, please provide the numbers in each age group and median age in each age group (not combined) in this first paragraph. Likewise, provide safety results (at least in a general statement) by age group.

Response 4: Thanks for your great suggestion. We have revised as your comments.

The numbers in each age group and median age in each age group (Page 12, line 244-246)

“In total, 119 (59.5%) participants were aged 18–59 years (median age 39; IQR, 34–45), and 81 (40.5%) participants were aged 60–80 years (median age 64; IQR, 63–69) in the study.”

Safety results by age group (Page 13, line 276-278)

“Participants 60–80 years of age had lower adverse reactions rates than those 18–59 years in both the inactivated vaccine (4.9% vs 39.0%, $P < 0.001$) and Ad5-nCoV (32.5% vs 70.0%, $P < 0.001$) groups.”

Point 5: Information on immunogenicity is provided by age group. It would be particularly interesting to know if safety differed by age group for the adenovirus boost, since immunogenicity did not.

Response 5: Thanks for your great suggestion. As your comments, our results showed that though the immunogenicity was no significant difference between 18-59 years and 60-80 years group after vaccinated with Ad5-nCoV, 60-80 years had lower adverse reactions rates than 18-59 years (32.5% vs 70.0%, $P < 0.001$), and logistic regression model of multivariate analysis showed 60-80 year was protective factor of adverse reaction (OR = 0.33). That could be a characteristic of Ad5-nCoV increased antibody levels, irrespective of age. For the lower adverse reactions rates of 60-80 years, in fact appeared not only participants vaccinated with Ad5-nCoV, but also inactivated vaccine and other vaccines. However, it is not clear what caused this phenomenon.

Point 6: Line 436 – “cloud” should be “could”.

Response 6: We are sorry for our carelessness. We have revised it in the manuscript.

Point 7: Line 445 – What is “cc”?

Response 7: We were really sorry for our spelling mistake and we have revised it.(Page 23, line 450-451) “Ad5-nCoV elicits a stronger humoral response and restores higher peak and more durable antibody levels than the inactivated SARS-CoV-2 vaccine.”

Point 8: Given the current state of the field, it would be useful for the authors to comment on the benefit of updating vaccines to contain more recent Covid-19 variants. Is that what they mean by “next generation” vaccines?

Response 8: We sincerely appreciate the valuable comments. The COVID-19 vaccines currently in development are very diverse, but all share a common goal - to provide durable protection against the virus's variants. However, We think the next generation vaccines was not only updating vaccines to contain more recent Covid-19 variants. They should have advantage to provide protection against undeveloped variants, while others provide stronger immunity, at lower doses, or better protection against infection and transmission of the virus.

We appreciate for Reviewers’ warm work earnestly, and hope the correction will meet with approval. Once again, thank you very much for your comments and suggestions. We would like to take this opportunity to thank you for all your time involved and this great opportunity for us to improve the manuscript.

Sincerely,

The Authors

Response to Reviewer 3's Comments

Point 1: Lines 278-297 – the addition of the GMFIs improves this section, however the GMT ratio (Geometric Mean Ratio [GMR]) between the arms (Ad5-nCoV/inactivated vaccine) at day 14 should be reported. The SAP states that the research hypothesis 1 is “the antibody level of Ad5-nCoV group 14 days after booster immunization is not inferior to that of ICV booster immunization group”; to answer the main hypothesis of the trial, the GMR of Ad5-nCoV/ICV at day 14 is what should be presented along with the 95% confidence interval. The lower bound of the confidence interval can then be assessed to determine if it meets the criteria to claim non-inferiority (>0.67).

Response 1: We sincerely appreciate the valuable comments. We have revised in the manuscript.(Page 14, line 302-304)

“The GMFI of the Ad5-nCoV/ inactivated vaccine was 9.5 (95% CI 7.0–12.9) on day 14, which met the criteria for non-inferiority (the lower bound of the confidence interval was >0.67).”

Point 2: Lines 404-406 – my original comment was perhaps unclear, so my apologies to the authors. I believe the authors wish to say that each of the factors (male, aged 60-80, chronic disease) *individually* are associated with lower frequency of adverse reactions, not necessarily in combination. If so, this sentence needs to be reworded.

Response 2: Thanks for your great suggestion on improving the accessibility of our manuscript. We have revised as your comments (Page 19, line 411-412).

“Interestingly, each of the factors (male, aged 60–80 years, and chronic diseases) is associated with a lower frequency of local and systemic adverse reactions.”

We would like to take this opportunity to thank you for all your time involved and this great opportunity for us to improve the manuscript. We look forward to hearing from you regarding our submission. We would be glad to respond to any further questions and comments that you may have.

Sincerely,

The Authors